# Measuring and Controlling Solution Degeneracy across Task-Trained Recurrent Neural Networks

**Ann Huang**[1,2,3], **Satpreet H. Singh**[2,3], **Flavio Martinelli**[2,3,4], **Kanaka Rajan**[2,3]

[1]Harvard University    [2]Harvard Medical School    [3]Kempner Institute    [4]EPFL

annhuang@g.harvard.edu

## Abstract

Task-trained recurrent neural networks (RNNs) are widely used in neuroscience and machine learning to model dynamical computations. To gain mechanistic insight into how neural systems solve tasks, prior work often reverse-engineers individual trained networks. However, different RNNs trained on the same task and achieving similar performance can exhibit strikingly different internal solutions, a phenomenon known as solution degeneracy. Here, we develop a unified framework to systematically quantify and control solution degeneracy across three levels: behavior, neural dynamics, and weight space. We apply this framework to 3,400 RNNs trained on four neuroscience-relevant tasks: flip-flop memory, sine wave generation, delayed discrimination, and path integration, while systematically varying task complexity, learning regime, network size, and regularization. We find that higher task complexity and stronger feature learning reduce degeneracy in neural dynamics but increase it in weight space, with mixed effects on behavior. In contrast, larger networks and structural regularization reduce degeneracy at all three levels. These findings empirically validate the Contravariance Principle and provide practical guidance for researchers seeking to tune the variability of RNN solutions, either to uncover shared neural mechanisms or to model the individual variability observed in biological systems. This work provides a principled framework for quantifying and controlling solution degeneracy in task-trained RNNs, offering new tools for building more interpretable and biologically grounded models of neural computation.

## 1   Introduction

Recurrent neural networks (RNNs) are widely used in machine learning and computational neuroscience to model dynamical processes. They are typically trained with standard nonconvex optimization methods and have proven useful as surrogate models for generating hypotheses about the neural mechanisms underlying task performance [1, 2, 3, 4, 5, 6]. Traditionally, the study of task-trained RNNs has focused on reverse-engineering a single trained model, implicitly assuming that networks trained on the same task would converge to similar solutions, even when initialized or trained differently. However, recent work has shown that this assumption does not hold universally, and the solution space of task-trained RNNs can be highly degenerate: networks may achieve the same level of training loss, yet differ in out-of-distribution (OOD) behavior, internal representations, neural dynamics, and connectivity [7, 8, 9, 10, 11, 12, 13]. For instance, [8] found that while trained RNNs may share certain topological features, their representational geometry can vary widely. Similarly, [7] showed that task-trained networks can develop qualitatively distinct neural dynamics and OOD generalization behaviors.

These findings raise fundamental questions about the solution space of task-trained RNNs: **What factors govern the solution degeneracy across independently trained RNNs?** When the solution space of task-trained RNNs is highly degenerate, to what extent can we trust conclusions drawn from a single model instance? While feedforward networks have been extensively studied in terms of how

weight initialization and stochastic training (e.g., mini-batch gradients) lead to divergent solutions, RNNs still lack a systematic and unified understanding of the factors that govern solution degeneracy [14, 15, 16, 17, 18, 19, 20, 21, 22, 23]. Cao and Yamins [24] proposed the *Contravariance Principle*, which posits that as the computational objective (i.e., the task) becomes more complex, the solution space should become less dispersed—since fewer models can simultaneously satisfy the stricter constraints imposed by harder tasks. While this principle is intuitive and compelling, it has thus far remained largely theoretical and has not been directly validated through empirical studies.

In this paper, we introduce a unified framework for quantifying solution degeneracy at three levels: behavior, neural dynamics, and weight space (Figure 1). Leveraging this framework, we isolate four key factors that control solution degeneracy: task complexity, learning regime, network width, and structural regularization. We apply this framework in a large-scale experiment, training 50 independently initialized RNNs on each of four neuroscience-relevant tasks. By systematically varying task complexity, learning regime, network width, and regularization, we map how each factor shapes degeneracy across behavior, dynamics, and weights. We find that as task complexity increases (whether via more input–output channels, higher memory demand, or auxiliary objectives, or as networks undergo stronger feature learning), their neural dynamics become more consistent, while their weight configurations

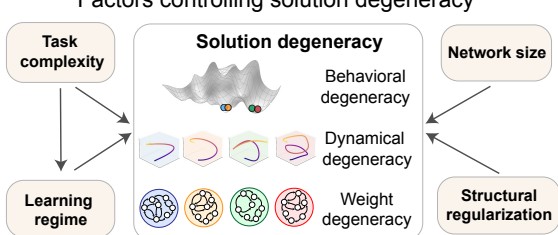

Figure 1: **Key factors shape degeneracy across behavior, dynamics, and weights.** Schematic of our framework for analyzing solution degeneracy in task-trained RNNs. We evaluate how task complexity, learning regime, network size, and structural regularization influence degeneracy at three levels: behavior (network outputs), neural dynamics (state trajectories), and weight space (connectivity).

grow more variable. In contrast, increasing network size or imposing structural regularization during training reduces variability at both the dynamics and weight levels. At the behavioral level, each of these factors reliably modulates behavioral degeneracy; however, the relationship between behavioral and dynamical degeneracy is not always consistent.

Table 1 summarizes how task complexity, learning regime, network size, and regularization affect degeneracy across levels. In both machine learning and neuroscience, the desired level of degeneracy may vary depending on the specific research questions being investigated. This framework offers practical guidance for tailoring training to a given goal, whether encouraging consistency across models [25], or promoting diversity across learned solutions [26, 27, 28].

**Our key contributions are as follows:**

- A unified framework for analyzing solution degeneracy in task-trained RNNs across behavior, dynamics, and weights.

- A systematic sweep of four factors: task complexity, feature learning, network size, and regularization, and a summary of their effects across levels (Table 1), with practical guidance for tuning consistency vs. diversity [25, 26, 27, 28].

- A double dissociation: task complexity and feature learning yield *contravariant* effects on weights vs. dynamics, while network size and regularization yield *covariant* effects. Here, contravariant means that a factor decreases degeneracy at one level (e.g., dynamics) while increasing it at another (e.g., weights), whereas covariant means both levels change in the same direction.

## 2 Methods

### 2.1 Model architecture and training procedure

We use discrete-time nonlinear *vanilla* recurrent neural networks (RNNs), defined by the update rule: $\mathbf{h}_t = \tanh\left(\mathbf{W}_h \mathbf{h}_{t-1} + \mathbf{W}_x \mathbf{x}_t + \mathbf{b}\right)$ where $\mathbf{h}_t \in \mathbb{R}^n$ is the hidden state, $\mathbf{x}_t \in \mathbb{R}^m$ is the input, $\mathbf{W}_h \in \mathbb{R}^{n \times n}$ and $\mathbf{W}_x \in \mathbb{R}^{n \times m}$ are the recurrent and input weight matrices, and $\mathbf{b} \in \mathbb{R}^n$ is a bias

vector. A learned linear readout is applied to the hidden state to produce the model's output at each time step. Networks are trained with Backpropagation Through Time (BPTT) [29], which unrolls the RNN over time to compute gradients at each step. All networks are trained using supervised learning with the Adam optimizer without weight decay. Learning rates are tuned per task (Appendix B). For each task, we train 50 RNNs with 128 hidden units. Weights are initialized from the uniform distribution $\mathcal{U}\left(-1/\sqrt{n},\, 1/\sqrt{n}\right)$ and hidden states are initialized to be zeros.

In all experiments, we train networks until them reach a near-asymptotic, task-specific mean-squared error (MSE) threshold on the training set (see Appendix B), after which we allow a patience period of 3 epochs and stop training to measure degeneracy. This early-stopping criterion ensures that networks trained on the same task achieve comparable final losses before any degeneracy analysis.

## 2.2 Task suite for diagnosing solution degeneracy

We selected a diverse set of four tasks designed to elicit distinct neural dynamics commonly studied in neuroscience. The **N-Bit Flip-Flop** task captures pattern recognition and memory retrieval processes, analogous to Hopfield-type attractor networks that store discrete binary patterns and retrieve them from partial cues [30, 31]. The **Delayed Discrimination** task models working memory maintenance in classic delayed-response paradigms [32, 33]. The **Sine Wave Generation** task represents pattern generation, analogous to Central Pattern Generators (CPGs) that produce self-sustaining rhythmic outputs underlying motor control [34], as well as oscillatory activity observed in motor cortex during movement [35]. Finally, the **Path Integration** task is inspired by hippocampal and entorhinal circuits that build a cognitive map of the environment to track position by integrating self-motion cues [36]. These tasks have also been used in prior benchmark suites for neuroscience-relevant RNN training [37, 38, 8], underscoring their broad relevance for studying diverse neural computations. Below, we briefly describe the task structure and the typical dynamics required to solve each one.

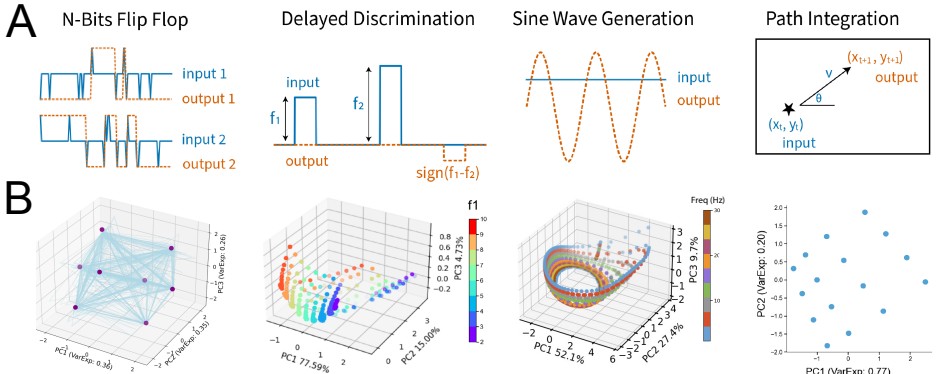

Figure 2: **Our task suite spans memory, integration, pattern generation, and decision-making.** Task schematics and representative network trajectories projected onto the top principal components are shown in (A)–(B). The four tasks are: **N-Bit Flip-Flop**: The network must remember the last nonzero input on each of $N$ independent channels. **Delayed Discrimination**: The network compares the magnitude of two pulses, separated by a variable delay, and outputs their sign difference. **Sine Wave Generation**: A static input specifies a target frequency, and the network generates the corresponding sine wave over time. **Path Integration**: The network integrates velocity inputs to track position in a bounded 2D or 3D arena (schematic shows 2D case).

**N-Bit Flip-Flop Task** Each RNN receives $N$ independent input channels taking values in $\{-1, 0, +1\}$, which switch with probability $p_{\text{switch}}$. The network has $N$ output channels that must retain the most recent nonzero input on their respective channels. The network dynamics form $2^N$ fixed points, corresponding to all binary combinations of $\{-1, +1\}^N$. The output range of this task is $[-1, 1]$ and we apply an early-stopping training MSE threshold at 0.001.

**Delayed Discrimination Task** The network receives two pulses of amplitudes $f_1, f_2 \in [2, 10]$, separated by a variable delay $t \in [5, 20]$ time steps, and must output $\text{sign}(f_2 - f_1)$. In the $N$-channel variant, comparisons are made independently across channels. The network forms task-relevant fixed

points to retain the amplitude of $f_1$ during the delay period. The output range of this task is $[-1, 1]$ and we apply an early-stopping training MSE threshold at 0.01.

**Sine Wave Generation** The network receives a static input specifying a target frequency $f \in [1, 30]$ and must generate the corresponding sine wave $\sin(2\pi f t)$ over time. We define $N_{\text{freq}}$ target frequencies, evenly spaced within the range $[1, 30]$, and use them during training. In the $N$-channel variant, each input channel specifies a frequency, and the corresponding output channel generates a sine wave at that frequency. For each frequency, the network dynamics form and traverse a limit cycle that produces the corresponding sine wave. The output range of this task is $[-1, 1]$ and we apply an early-stopping training MSE threshold at 0.05.

**Path Integration Task** Starting from a random position in 2D, the network receives angular direction $\theta$ and speed $v$ at each time step and updates its position estimate. In the 3D variant, the network takes as input azimuth $\theta$, elevation $\phi$, and speed $v$, and outputs updated $(x, y, z)$ position. The network performs path integration by accumulating velocity vectors based on the input directions and speeds. After training, the network forms a map of the environment in its internal state space. The output range of this task is $[-5, 5]$ and we apply an early-stopping training MSE threshold at 0.05.

In our task suite, trained RNNs develop distinct stable dynamical objects: fixed-point (N-Bit Flip Flop, Delayed Discrimination), limit cycle (Sine Wave Generation), and attractor manifold (Path Integration). In Appendix E, we extend our task suite to include a next-step prediction task on the Lorenz 96 chaotic attractors [39], where networks exhibit chaotic dynamical regime.

### 2.3 Multi-level framework for quantifying degeneracy

#### 2.3.1 Behavioral degeneracy

We define a novel metric for behavioral degeneracy as the variability in network responses to out-of-distribution (OOD) inputs. We quantify OOD performance as the mean squared error of all converged networks that achieved near-asymptotic training loss under a *temporal generalization* condition. For the Delayed Discrimination task, we doubled the delay period. For all other tasks, we doubled the length of the entire trial to assess generalization under extended temporal contexts. Behavioral degeneracy is defined as standard deviation of the OOD losses: $\sigma_{\text{OOD}} = \sqrt{\frac{1}{N} \sum_{i=1}^{N} \left( \mathcal{L}_{\text{OOD}}^{(i)} - \overline{\mathcal{L}}_{\text{OOD}} \right)^2}$,

where $\overline{\mathcal{L}}_{\text{OOD}}$ is the mean OOD loss. While we focus primarily on the *temporal generalization* condition for behavioral degeneracy since it directly probes RNNs' sequence processing capacities and their ability to generalize across extended temporal horizons, the same metric can be readily applied to other OOD conditions, such as input noise or external perturbations. In the rest of the paper, we use the term *behavioral degeneracy [temporal generalization]* to explicitly indicate the OOD condition being tested.

#### 2.3.2 Dynamical degeneracy

We use Dynamical Similarity Analysis (DSA) [40] to compare the neural dynamics of task-trained networks through pairwise analyses. While previous comparison methods mostly focus on geometry of the data [41, 42, 43, 44], RNNs implement computations through time-varying trajectories rather than static representations, and two RNNs exhibiting similar representational geometry can implement distinct dynamical computations, and vise versa. DSA compares the topological structure of the neural dynamics and has been shown to be more robust to noise and better at identifying behaviorally relevant differences than geometry-based comparison method [45]. For a pair of networks $X$ and $Y$, DSA projects their time series of activities to a higher-dimensional space and identifies a linear dynamic operator for each system via next-step prediction. The DSA distance between two systems is then computed by minimizing the Frobenius norm between the operators, up to an orthogonal transformation (rotation and reflection):

$$d_{\text{DSA}}(A_x, A_y) = \min_{C \in O(n)} \left\| A_x - C A_y C^{-1} \right\|_F,$$

where $O(n)$ is the orthogonal group. We define dynamical degeneracy as the average DSA distance across all network pairs. Additional details on the DSA metric are provided in Appendix F. We note that scale of the DSA distance used to quantify dynamical degeneracy can depend on the choice of DSA hyperparameters. To ensure fair comparison across conditions, we keep all DSA

hyperparameters fixed for RNNs trained on the same task. To assess if the neural dynamics across different trained networks are statistically different, we also establish a null distribution by comparing neural trajectories sampled from the same underlying network, see Appendix F.3 for details.

We focus on comparing neural dynamics because RNNs implement computations through time-evolving trajectories rather than static input representations. In addition, we assess representational degeneracy using Singular Vector Canonical Correlation Analysis (SVCCA) [41]. As shown in Appendix G, the four factors that influence dynamical degeneracy do not impose the same constraints on representational degeneracy.

### 2.3.3 Weight degeneracy

We quantify weight-level degeneracy via a permutation-invariant version of the Frobenius norm, defined as:
$$d_{\mathrm{PIF}}(\mathbf{W}_1, \mathbf{W}_2) = \min_{\mathbf{P} \in \mathcal{P}(n)} \left\| \mathbf{W}_1 - \mathbf{P}^\top \mathbf{W}_2 \mathbf{P} \right\|_F$$

where $\mathbf{W}_1$ and $\mathbf{W}_2$ are the recurrent weight matrices for a pair of RNNs, $\mathcal{P}(n)$ is the set of permutation matrices of size $n \times n$, and $\| \cdot \|_F$ denotes the Frobenius distance. See Appendix F.2 for additional details. For comparing $d_{PIF}$ computed on networks of different sizes, we normalize the above norm by the number of parameters in the weight matrix.

## 3 Results

### 3.1 Task complexity modulates degeneracy across levels

To investigate how task complexity influences dynamical degeneracy, we varied the number of independent input–output channels. This increased the representational load by forcing networks to solve multiple input-output mappings simultaneously. To visualize how neural dynamics vary across networks, we applied two-dimensional Multidimensional Scaling (MDS) to their pairwise distances. As task complexity increased, network dynamics became more similar, forming tighter clusters in the MDS space (Figure 3A). This contravariant relationship between task complexity and dynamical degeneracy was consistent across all tasks (Figure 3B). Higher task demands constrain the space of viable dynamical solutions, leading to greater consistency across independently trained networks.

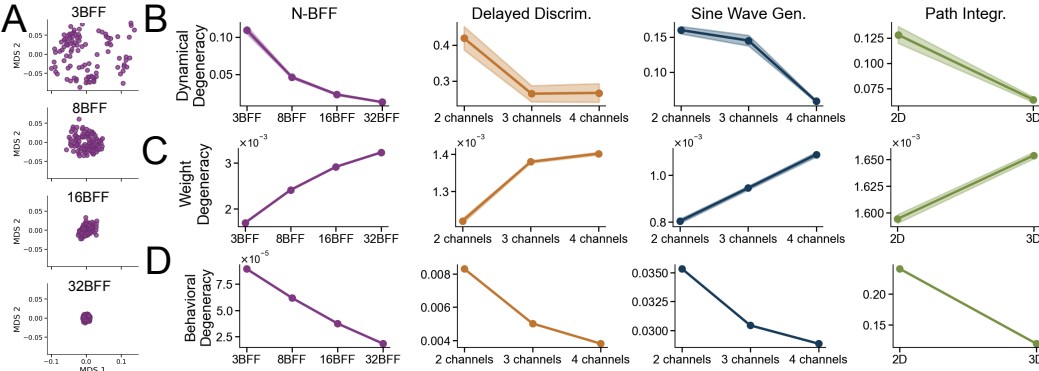

Figure 3: **Higher task complexity reduces dynamical and behavioral degeneracy, but increases weight degeneracy.** **(A)** Two-dimensional MDS embedding of network dynamics shows that independently trained networks converge to more similar trajectories as task complexity increases. **(B)** Dynamical, **(C)** weight, and **(D)** behavioral degeneracy [temporal generalization] across 50 networks as a function of task complexity. Shaded area indicates $\pm 1$ standard error.

At the behavioral level, networks trained on more complex tasks consistently showed lower variability in their responses to OOD test inputs (Figure 3D) in the temporal generalization condition. This finding suggests that increased task complexity, by reducing dynamical degeneracy, also leads to more consistent and less degenerate behavior on the temporal generalization condition across networks. Together, the results at the behavioral and dynamical levels support the *Contravariance Principle*, which posits an inverse relationship between task complexity and the dispersion of network solutions [24].

At the weight level, we found that pairwise distances between converged RNNs' weight matrices increased consistently with task complexity (Figure 3C). This likely reflects increased dispersion of local minima in weight space for harder tasks. This interpretation is consistent with prior work on mode averaging and loss landscape geometry in feedforward networks, showing that harder tasks tend to yield increasingly isolated minima, separated by steeper barriers [46, 47, 48, 49, 50, 51, 52]. A complementary perspective comes from [53] who introduced the *intrinsic dimension* as the lowest-dimensional weight subspace that still contains a solution, which can serve as a proxy for task complexity. As task complexity increases, the intrinsic dimension of the weight space expands and each solution occupies a thinner slice of a higher-dimensional space, leading to minima that lie further apart. In Section 3.2, we propose an additional mechanism: an interaction between task complexity and the network's learning regime that further amplifies weight-space degeneracy.

### 3.1.1 Additional axes of task complexity

In earlier experiments, we controlled task complexity by varying the number of independent input–output channels, effectively duplicating the task across dimensions. Here, we explore two alternative approaches: increasing the task's memory demand and adding auxiliary objectives.

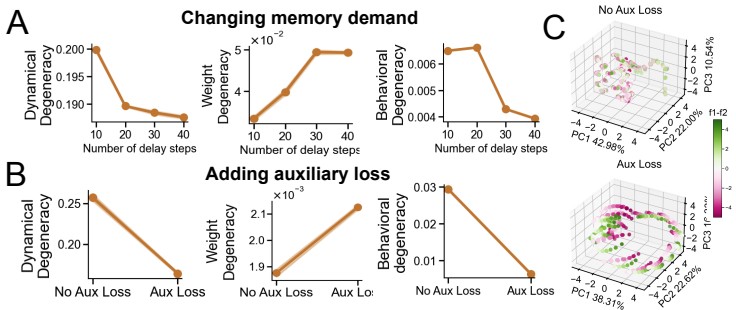

**Changing memory demand.** Of the four tasks, only Delayed Discrimination requires extended memory, as its performance depends on maintaining the first stimulus across a variable delay.

Figure 4: **Increasing memory demand or adding auxiliary loss changes task complexity, which in turn modulates degeneracy.** In the Delayed Discrimination task, both manipulations reduce dynamical and behavioral degeneracy [temporal generalization] while increasing weight degeneracy. The auxiliary loss also induces additional line attractors in the network's dynamics, as shown in (C).

See Appendix D for a quantification of each task's memory demand. We increased the memory load in Delayed Discrimination by lengthening the delay period. This manipulation reduced degeneracy at the dynamical and behavioral levels but increased it at the weight level, mirroring the effect of increasing task dimensionality (Figure 4A).

**Adding auxiliary loss.** We next examined how adding an auxiliary loss affects solution degeneracy in the Delayed Discrimination task. Specifically, the network outputs both the sign and the magnitude of the difference between two stimulus values ($f_2 - f_1$), using separate output channels for each. This manipulation added a second output channel and increased memory demand by requiring the network to track the magnitude of the difference between incoming stimuli. Consistent with our hypothesis, this manipulation reduced dynamical and behavioral degeneracy [temporal generalization] while increasing weight degeneracy (Figure 4B). Crucially, the auxiliary loss induced additional line attractors in the network dynamics, further structuring internal trajectories and aligning neural responses across networks (Figure 4C). While the auxiliary loss increases both output dimensionality and temporal memory demand, we interpret its effect holistically as a structured increase in task complexity.

## 3.2 Feature learning

### 3.2.1 Task complexity scales feature learning

In deep learning theory, neural networks can either solve tasks using their random features at initialization, or adapt their weights and internal features to capture task specific structure [54, 55, 56, 57]. These are referred to as the *lazy learning* regime, where weights and internal features remain largely unchanged during training, and the *rich learning*, or *feature learning* regime, where networks reshape their hidden representations and weights to capture task-specific structure [54, 58, 59, 55]. As the complexity of a task grows, the initial random features no longer suffice to solve it, pushing the network beyond the lazy regime and into feature learning, where weights and internal representations

adapt more substantially. [60, 61]. If more complex task variants, like those in Section 3.1, truly induce greater feature learning, then networks should adapt more from their initializations and traverse a greater distance in the weight space, resulting in more dispersed final weights.

We therefore hypothesize that the increased weight degeneracy observed in harder tasks reflects stronger feature learning within the network. To test this idea, we measured feature learning strength in networks trained on different task variants using two complementary metrics [62, 58]: **Weight-change norm:** $\|\mathbf{W}_T - \mathbf{W}_0\|_F$, where larger values indicate stronger feature learning. **Kernel alignment (KA):** The geometry of learning under gradient descent can be described by the neural tangent kernel (NTK), which captures how weight updates affect the network outputs. The

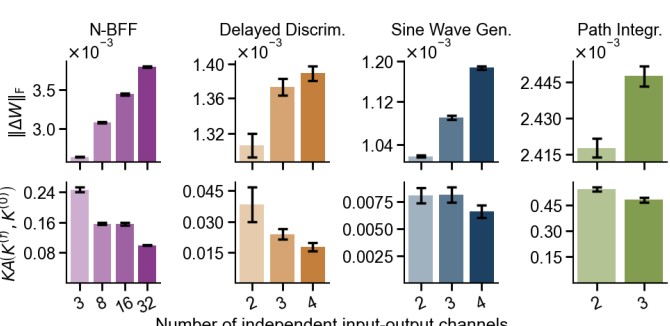

Figure 5: **More complex tasks drive stronger feature learning in RNNs.** Increased input–output dimensionality leads to higher weight-change norms ($\|\Delta W\|_F$) and lower kernel alignment (KA). Error bars indicate $\pm 1$ standard error.

NTK is defined by $K = \nabla_W \hat{y}^\top \nabla_W \hat{y}$ where $\hat{y}$ denotes the network output. KA measures the directional change of the NTK before and after training: $\mathrm{KA}\left(K^{(T)}, K^{(0)}\right) = \frac{\mathrm{Tr}\left(K^{(T)} K^{(0)}\right)}{\left\|K^{(T)}\right\|_F \left\|K^{(0)}\right\|_F}$. Lower KA indicates greater NTK rotation and thus stronger feature learning.

We find that more complex tasks consistently drive stronger feature learning and greater dispersion in weight space, as reflected by increasing weight-change norm and decreasing kernel alignment across all tasks (Figure 5).

### 3.2.2 Controlling feature learning reshapes degeneracy across levels

Our earlier results show that harder tasks induce stronger feature learning, which in turn shapes the dispersion of solutions in the weight space. To test whether feature learning *causally* affects degeneracy, we used a principled network parameterization known as maximum update parameterization ($\mu P$), which allows stable feature learning across network widths, even in the infinite-width limit [57, 54, 56, 55]. In this setup, a single hyperparameter ($\gamma$) controls the strength of feature learning: higher $\gamma$ values induce a richer feature-learning regime. Under this parameterization, the network update rule, initialization, and learning rate are scaled with respect to network width $N$. For the Adam optimizer, the output is scaled as $f(t) = \frac{1}{\gamma N} W_{\mathrm{readout}} \phi(h(t))$. The hidden state update is scaled as $h(t+1) - h(t) = \tau\left(-h(t) + \frac{1}{N} J\phi(h(t)) + Ux(t)\right)$, where $J_{ij} \sim \mathcal{N}(0, N)$ are the recurrent weights and $\phi$ is the *tanh* nonlinearity. The learning rate scales as $\eta = \gamma \eta_0$. A detailed explanation of $\mu P$ and its relationship to the standard parameterization is in Appendix K and L. For each task, we trained networks with multiple $\gamma$ values and confirmed that larger $\gamma$ consistently induces stronger feature learning, as evidenced by increased weight-change norm and decreased kernel alignment (Appendix M).

We observed that stronger feature learning reduced degeneracy at the dynamical level but increased it at the weight level. We see that when $\gamma$ is high, networks tend to learn similar task-specific features and converge to consistent dynamics and behavior. In contrast, lazy networks (with small $\gamma$) rely on their initial random features, leading to more divergent solutions across seeds—even though their weights move less overall (Figure 6). This finding aligns with prior work in feedforward networks, where feature learning was shown to reduce the variance of the neural tangent kernel across converged models [60]. At the behavioral level, however, increasing feature-learning strength leads networks to overfit the training distribution (Appendix J.2). We hypothesize that stronger feature learning exacerbates overfitting, increasing both average OOD loss and the variability of OOD behavior across models (Figure 6) [63, 64, 65, 66]. Although stronger feature learning increases behavioral degeneracy [temporal generalization], this may partially reflect overfitting to the training distribution,

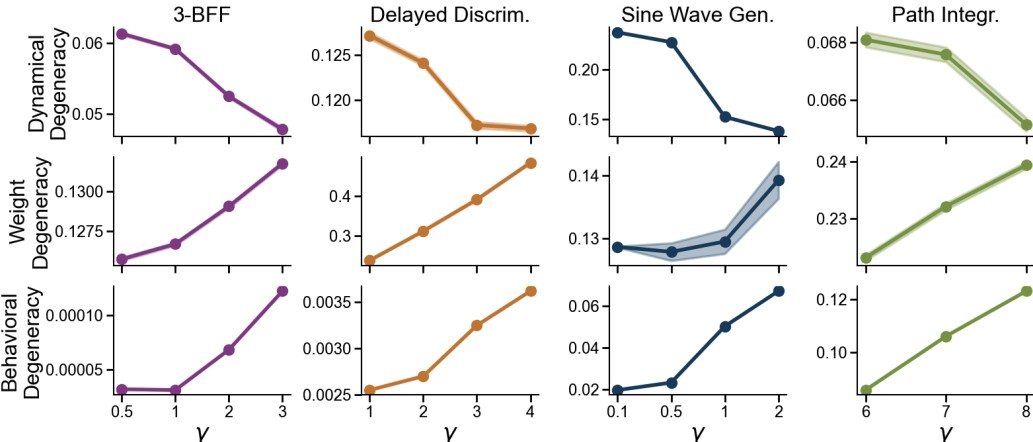

Figure 6: **Stronger feature learning reduces dynamical degeneracy but increases weight and behavioral degeneracy.** Panels show degeneracy at the dynamical, weight, and behavioral levels (top to bottom). Shaded area indicates $\pm 1$ standard error.

an effect we highlight in Appendix J.2. Clarifying the mechanistic link between dynamical and behavioral degeneracy [temporal generalization] remains an important direction for future work. In Appendix I, we demonstrate that the observed effects of feature learning on degeneracy both interpolates smoothly within the range of $\gamma$ values and extrapolates beyond the range reported in Figure 6.

## 3.3 Larger networks yield more consistent solutions across levels

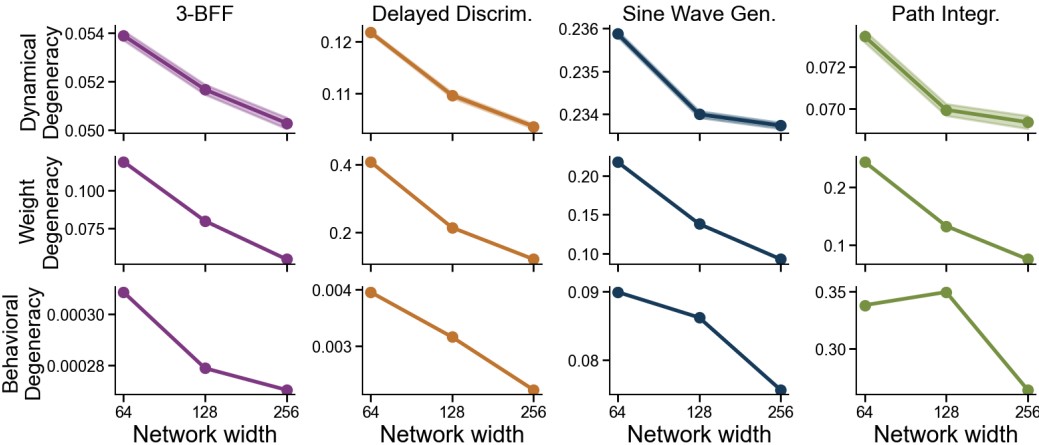

Figure 7: **Larger networks reduce degeneracy across weight, dynamics, and behavior.** After controlling for feature learning strength ($\gamma = 1$ held constant across network widths), wider RNNs yield more consistent solutions across all three levels of analysis. Panels show degeneracy at the dynamical, weight, and behavioral levels (top to bottom). Shaded area indicates $\pm 1$ standard error.

Prior work in machine learning and optimization shows that over-parameterization improves convergence by helping gradient methods escape saddle points [67, 68, 69, 70, 71, 72, 16]. We therefore hypothesized that larger RNNs would converge to more consistent solutions across seeds. However, increasing width also tends to push models towards the lazy regime, where feature learning is suppressed [73, 59, 54, 55, 56]. To disentangle these competing effects, we again use the $\mu P$ parameterization, which holds feature learning strength constant (via fixed $\gamma$) while scaling width. Although larger networks may yield more consistent solutions via self-averaging, this outcome is not guaranteed without controlling for feature learning. In standard RNNs, increasing width often

Table 1: Summary of how each factor affects solution degeneracy. Arrows indicate the direction of change for each level as the factor increases. Contravariant factors shift **dynamic** and **weight degeneracy** in opposite direction; covariant factors shift them in the same directions.

| Factor | Dynamics | Weights | Behavior |
|---|---|---|---|
| **Higher Task complexity (contravariant)** | ↓ | ↑ | ↓ |
| **More Feature learning (contravariant)** | ↓ | ↑ | ↑ |
| **Larger Network size (covariant)** | ↓ | ↓ | ↓ |
| **Regularization (covariant)** | ↓ | ↓ | ↓ |

induces lazier dynamics, which can paradoxically increase dynamical degeneracy rather than reduce it. The $\mu P$ setup enables us to isolate the size effect cleanly.

Across all tasks, larger networks consistently exhibit lower degeneracy at the weight, dynamical, and behavioral levels, producing more consistent solutions across random seeds (Figure 7). Our dense sweep over 12 intermediate network sizes from 32 to 512 on the 3-Bits Flip Flop task in Appendix I further confirms the observed effect of network width on degeneracy. This pattern aligns with findings in vision and language models, where wider networks converge to more similar internal representations [74, 75, 41, 76, 77, 65]. In recurrent networks, only a few studies have investigated this "convergence-with-scale" effect using representation-based metrics [74, 78]. Our results extend these findings by (1) focusing on neural computations across time (i.e., neural dynamics) rather than static representations, and (2) demonstrating convergence-with-scale across weight, dynamical, and behavioral levels in RNNs.

### 3.4 Structural regularization reduces solution degeneracy

Low-rank and sparsity constraints are widely used structural regularizers in neuroscience-inspired modeling and efficient machine learning [4, 79, 80, 81, 82]. A low-rank penalty compresses the weight matrices into a few dominant modes, while an $\ell_1$ penalty drives many parameters to zero and induces sparsity. In both cases, task-irrelevant features are pruned, nudging independently initialized networks toward more consistent solutions on the same task. To test this idea, we augmented the task loss with either a nuclear-norm penalty on the recurrent weights $\mathcal{L} = \mathcal{L}_{\text{task}} + \lambda_{\text{rank}} \sum_{i=1}^{r} \sigma_i$, where $\sigma_i$ are the singular values of the recurrent matrix, or an $\ell_1$ sparsity penalty: $\mathcal{L} = \mathcal{L}_{\text{task}} + \lambda_{\ell_1} \sum_i |w_i|$. We focused on the Delayed Discrimination task to control for baseline difficulty, and observe that both regularizers consistently reduced degeneracy across all levels. Similar effects hold in other tasks (Appendix O, Figure 8) and intermediate regularization strengths (Appendix I).

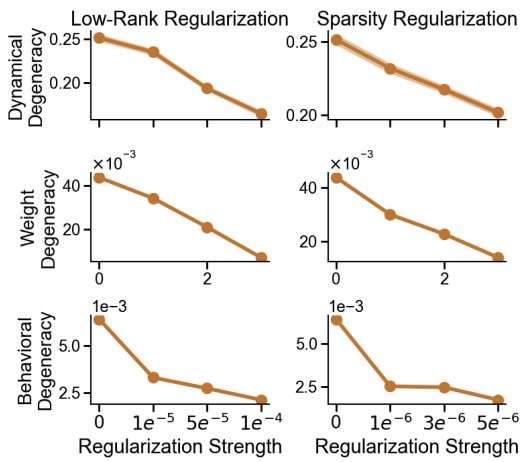

Figure 8: **Low-rank and sparsity regularization reduce solution degeneracy across all levels.** On the Delayed Discrimination task, both regularizers lower degeneracy in dynamics, weights, and behavior. Shaded area indicates ±1 standard error.

## 4 Discussion

In this work, we introduced a unified framework for quantifying solution degeneracy in task-trained recurrent neural networks (RNNs) at three complementary levels: behavior, neural dynamics, and weights. We systematically varied four factors within our generalizable framework: (i) task complexity (via input–output dimensionality, memory demand, or auxiliary loss), (ii) feature learning strength, (iii) network size, and (iv) structural regularization. We then evaluated their effects on solution degeneracy across a diverse set of neuroscience-relevant tasks.

Two consistent patterns emerged from this analysis. First, increasing task complexity or boosting feature learning produced a **contravariant** effect: dynamical degeneracy decreased while weight degeneracy increased. Second, increasing network size or applying structural regularization reduced degeneracy at both the weight and dynamical levels—that is, a **covariant** effect. Here, covariant and contravariant refer to the relationship between weight and dynamic degeneracy, not whether degeneracy increases or decreases overall. For example, task complexity and feature learning reduce dynamical degeneracy but increase weight degeneracy, whereas network size and regularization reduce both.

We also observed that the relationship between dynamical and behavioral degeneracy depends on the varying factor. For instance, stronger feature learning leads to more consistent neural dynamics on the training task but greater variability in OOD generalization This suggests that tightly constrained dynamics on the training set do not guarantee more consistent behavior on OOD inputs. This highlights the need for further empirical and theoretical work on how generalization depends on the internal structure of task-trained networks [83, 84, 85]. This divergence highlights a key open question: how much of behavioral consistency generalizes beyond training-aligned dynamics, and what task or network factors drive this decoupling?

These knobs allow researchers to tune the level of degeneracy in task-trained RNNs to suit specific research questions or application needs. For example, researchers may want to suppress degeneracy to study **common mechanisms** underlying a neural computation. Conversely, to probe **individual differences**, they can increase degeneracy to expose solution diversity across independently trained networks [86, 87, 88, 89, 90]. Our framework also supports ensemble-based modeling of brain data. By comparing dynamical and behavioral degeneracy across trained networks, it may be possible to match inter-individual variability in models to that observed in animals, helping capture the full distribution of task-solving strategies [91, 92, 93, 94].

Although our analyses use artificial networks, several of the mechanisms we uncover may translate directly to experimental neuroscience. For example, introducing an auxiliary sub-task during behavioral shaping, which mirrors our auxiliary-loss manipulation, could constrain the solution space animals explore, thereby reducing behavioral degeneracy [95]. Finally, our contrasting findings motivate theoretical analysis, e.g., using linear RNNs to understand why some factors induce contravariant versus covariant relationships across behavioral, dynamical, and weight-level degeneracy.

In summary, our work takes a first step toward addressing this classic puzzle in task-driven modeling: What factors shape the variability across independently trained networks? We present a unified framework for quantifying solution degeneracy in task-trained RNNs, identify the key factors that shape the solution landscape, and provide practical guidance for controlling degeneracy to match specific research goals in neuroscience and machine learning.

**Limitations and future directions.** This work considers networks equivalent if they achieve similar training loss. Future work could extend the framework to tasks with multiple qualitatively distinct solutions, to examine whether specific factors bias the distribution of networks across those solutions. Another open question is the observed decoupling between dynamical and behavioral degeneracy: how much of behavioral consistency generalizes beyond training-aligned dynamics, and what task or network factors drive this divergence.

# 5   Acknowledgments

We acknowledge funding from NIH (RF1DA056403, U01NS136507), James S. McDonnell Foundation (220020466), Simons Foundation (Pilot Extension-00003332-02, McKnight Endowment Fund, CIFAR Azrieli Global Scholar Program, NSF (2046583), Harvard Medical School Neurobiology Lefler Small Grant Award, Harvard Medical School Dean's Innovation Award, Alice and Joseph Brooks Fund Postdoctoral Fellowship, and Kempner Graduate Fellowship. This work has been made possible in part by a gift from the Chan Zuckerberg Initiative Foundation to establish the Kempner Institute for the Study of Natural and Artificial Intelligence at Harvard University.

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

# Appendix

## A    Task details

### A.1    N-Bit Flip Flop

| Task Parameter | Value |
|---|---|
| Probability of flip | 0.3 |
| Number of time steps | 100 |

### A.2    Delayed Discrimination

| Task Parameter | Value |
|---|---|
| Number of time steps | 60 |
| Max delay | 20 |
| Lowest stimulus value | 2 |
| Highest stimulus value | 10 |

### A.3    Sine Wave Generation

| Task Parameter | Value |
|---|---|
| Number of time steps | 100 |
| Time step size | 0.01 |
| Lowest frequency | 1 |
| Highest frequency | 30 |
| Number of frequencies | 100 |

### A.4    Path Integration

| Task Parameter | Value |
|---|---|
| Number of time steps | 100 |
| Maximum speed ($v_{max}$) | 0.4 |
| Direction increment std ($\theta_{std}$ / $\phi_{std}$) | $\pi/10$ |
| Speed increment std | 0.1 |
| Noise std | 0.0001 |
| Mean stop duration | 30 |
| Mean go duration | 50 |
| Environment size (per side) | 10 |

# B    Training details

## B.1   N-Bit Flip Flop

| Training Hyperparameter | Value |
| --- | --- |
| Optimizer | Adam |
| Learning rate | 0.001 |
| Learning rate scheduler | None |
| Max epochs | 300 |
| Steps per epoch | 128 |
| Batch size | 256 |
| Early stopping threshold | 0.001 |
| Patience | 3 |
| Time constant ($\mu P$) | 1 |

## B.2   Delayed Discrimination

| Training Hyperparameter | Value |
| --- | --- |
| Optimizer | Adam |
| Learning rate | 0.001 |
| Learning rate scheduler | CosineAnnealingWarmRestarts |
| Max epochs | 500 |
| Steps per epoch | 128 |
| Batch size | 256 |
| Early stopping threshold | 0.01 |
| Patience | 3 |
| Time constant ($\mu P$) | 0.1 |

## B.3   Sine Wave Generation

| Training Hyperparameter | Value |
| --- | --- |
| Optimizer | Adam |
| Learning rate | 0.0005 |
| Learning rate scheduler | None |
| Max epochs | 500 |
| Steps per epoch | 128 |
| Batch size | 32 |
| Early stopping threshold | 0.05 |
| Patience | 3 |
| Time constant ($\mu P$) | 1 |

### B.4  Path Integration

| Training Hyperparameter | Value |
|---|---|
| Optimizer | Adam |
| Learning rate | 0.001 |
| Learning rate scheduler | ReduceLROnPlateau |
| Learning rate decay factor | 0.5 |
| Learning rate decay patience | 40 |
| Max epochs | 1000 |
| Steps per epoch | 128 |
| Batch size | 64 |
| Early stopping threshold | 0.05 |
| Patience | 3 |
| Time constant ($\mu P$) | 0.1 |

## C  Task performance of trained networks

In all experiments, we train networks until them reach a **near-asymptotic, task-specific mean-squred error (MSE) threshold** (0.001 for N-BFF, 0.01 for Delayed Discrimination, and 0.05 for Sine-Wave Generation and Path Integration), after which we allow a patience period of 3 epochs and stop training to measure degeneracy. This early-stopping criterion ensures that networks trained on the same task/condition achieve comparable final losses before any degeneracy analysis.

To quantify the residual variation, we report the coefficient of variation (CV) of the final training loss across seeds for each condition, expressed as % of the mean. Header labels match the x-axis levels used in the main-text figures. Final losses cluster tightly near small values of the loss threshold, so even a double-digit CV translates to very small absolute variation. For example, a 10% CV at an MSE of 0.001 implies an s.d. of $10^{-4}$; at 0.01 it's $10^{-3}$. Additionally, the networks converged well on a global scale. Across our experiments, the mean MSE after training is under 2% of the mean MSE at initialization, indicating that training has converged well. Individual values: 0.059% (N-BFF), 1.6% (Delayed Discrimination), 0.32% (Sine-Wave Generation), 0.94% (Path Integration). CV can look large when the mean is tiny (the denominator is small). For example, a 16% CV on Sine-Wave Generation task corresponds to $\tilde{0}.05\%$ of the initialization loss, which is consistent with minor differences due to the stochastic gradients rather than under-training.

These variability values are also not monotonic in any factor and sometimes move opposite to the degeneracy trends, arguing against a loss-dispersion confound to solution degeneracy.

Table 2: Coefficient of variation (CV) of the final training loss across 50 networks for each task complexity level.

| Task Complexity | Level 1 | Level 2 | Level 3 | Level 4 |
|---|---|---|---|---|
| N-BFF | 6.30% | 4.60% | 9.30% | 3.50% |
| Delayed Discrim. | 15.90% | 8.40% | 9.50% | — |
| Sine Wave Gen. | 9.94% | 9.20% | 8.70% | — |
| Path Integr. | 9.16% | 2.85% | — | — |

Table 3: Coefficient of variation (CV) of the final training loss across 50 networks for each feature learning strength ($\gamma$)

| Feature Learning Strength | $\gamma_1$ | $\gamma_2$ | $\gamma_3$ | $\gamma_4$ |
|---|---|---|---|---|
| N-BFF | 9.70% | 9.10% | 13.40% | 11.70% |
| Delayed Discrim. | 8.70% | 12.60% | 11.70% | 12.30% |
| Sine Wave Gen. | 3.50% | 3.90% | 10.90% | 11.70% |
| Path Integr. | 5.40% | 5.20% | 6.20% | — |

Table 4: Coefficient of variation (CV) of the final training loss across 50 networks for each network width.

| Network Width | 64 units | 128 units | 256 units |
|---|---|---|---|
| N-BFF | 3.80% | 4.20% | 3.50% |
| Delayed Discrim. | 3.30% | 3.00% | 3.20% |
| Sine Wave Gen. | 17.80% | 16.60% | 16.40% |
| Path Integr. | 5.10% | 5.40% | 5.90% |

Table 5: Coefficient of variation (CV) of the final training loss across 50 networks for each L1 regularization strength.

| L1 Regularization | $\lambda_1$ | $\lambda_2$ | $\lambda_3$ | $\lambda_4$ |
|---|---|---|---|---|
| N-BFF | 2.10% | 6.90% | 1.10% | — |
| Delayed Discrim. | 15.90% | 14.50% | 16.70% | 14.90% |
| Sine Wave Gen. | 10.40% | 11.10% | 11.10% | — |
| Path Integr. | 9.00% | 7.10% | 3.00% | — |

Table 6: Coefficient of variation (CV) of the final training loss across 50 networks for each rank regularization strength.

| Rank Regularization | $\lambda_1$ | $\lambda_2$ | $\lambda_3$ | $\lambda_4$ |
|---|---|---|---|---|
| N-BFF | 2.10% | 7.20% | 4.30% | — |
| Delayed Discrim. | 15.90% | 16.90% | 13.60% | 12.10% |
| Sine Wave Gen. | 13.90% | 14.30% | 15.90% | — |
| Path Integr. | 7.70% | 7.90% | 6.70% | — |

## D   Memory demand of each task

In this section, we quantify each task's memory demand by measuring how far back in time its inputs influence the next output. Specifically, for each candidate history length $h$, we build feature vectors

$$\mathbf{s}_t^{(h)} = [\, x_{t-h+1}, \,\ldots,\, x_t;\; y_t \,] \; \in \; \mathbb{R}^{h\,d_{\text{in}}+d_{\text{out}}},$$

and **train a two-layer MLP to predict the subsequent target** $y_{t+1}$. We then evaluate the held-out mean-squared error $\text{MSE}(h)$, averaged over multiple random initializations. We identify the smallest history length $h^*$ at which the error curve plateaus or has a minimum, and take $h^*$ as the task's intrinsic memory demand.

From the results, we can see that the N-Bits Flip-Flop task requires only one time-step of memory—exactly what's needed to recall the most recent nonzero input in each channel. The Sine Wave Generation task demands two time-steps, reflecting the need to track both phase and direction of change. Path Integration likewise only needs one time-step, since the current position plus instantaneous velocity and heading suffice to predict the next position. Delayed Discrimination is the only memory-intensive task: our method estimates a memory demand of 25 time-steps, which happens to be the time interval between the offset of the first stimulus and the onset of the response period, during which the network needs to first keep track of the amplitude of the first stimulus and then its decision.

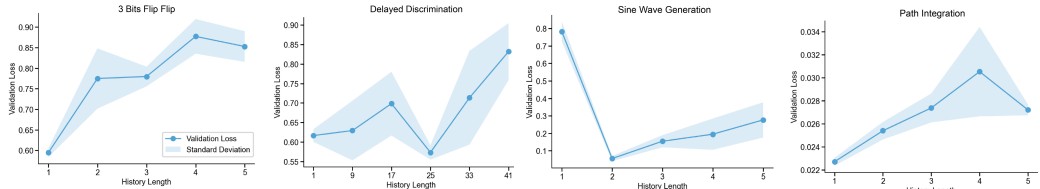

Figure 9: **Memory demand of each task**. The held-out mean-squared error $\mathrm{MSE}(h)$ of a two-layer MLP predictor is plotted against history length $h$. The intrinsic memory demand $h^*$, defined by the plateau or minimum of each curve, is 1 for the N-Bits Flip-Flop and Path Integration tasks, 2 for Sine Wave Generation, and 25 for Delayed Discrimination—matching the inter-stimulus delay interval in that task.

# E    Solution degeneracy in chaotic RNNs

Our original task suite comprises neuroscience-motivated tasks that produce stable-attractors: fixed-point (N-Bit Flip Flop, Delayed Discrimination), limit cycle (Sine Wave Generation), and attractor manifold (Path Integration). To further demonstrate that the observed effect of the four factors on degeneracy extend to RNNs with chaotic activity, here we add a chaotic attractor task and verified that the effects of all four factors on dynamical and weight degeneracy are consistent with Table 1.

**Lorenz 96 Attractor Dataset**    We simulated trajectories from the Lorenz 96 dynamical system [39], defined by

$$\frac{dx_i}{dt} = (x_{i+1} - x_{i-2})x_{i-1} - x_i + F, \quad i = 1, \ldots, N,$$

with cyclic boundary conditions $x_{-1} = x_{N-1}$, $x_0 = x_N$, $x_{N+1} = x_1$. The external forcing parameter was set to $F = 8.0$, a standard choice that induces chaotic dynamics.

To generate the dataset, we numerically integrated the system for $N = 16, 24$, and 32 dimensions. Each simulation used a time step of $\Delta t = 0.01$ and produced 15000 time points after discarding an initial transient of 1000 steps to remove non-stationary behavior. Initial conditions were sampled as small random perturbations around the fixed point $x_i = F$:

$$x_i(0) = F + 0.1\,\varepsilon_i, \quad \varepsilon_i \sim \mathcal{N}(0, 1).$$

For each condition, we trained 50 RNNs on next-step prediction until the networks achieve a near-asymptotic MSE loss at $0.0005$. After training, the average Lyapunov exponent of RNNs trained on the Lorenz 96 attractor with 16 dimensions is $12.58 \pm 0.74$, indicating chaotic neural dynamics.

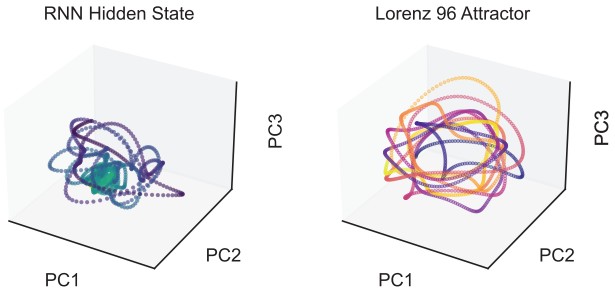

Figure 10: RNN recurrent activities and Lorenz 96 attractor ($N = 16$) trajectories projected onto their respective top 3 principle components.

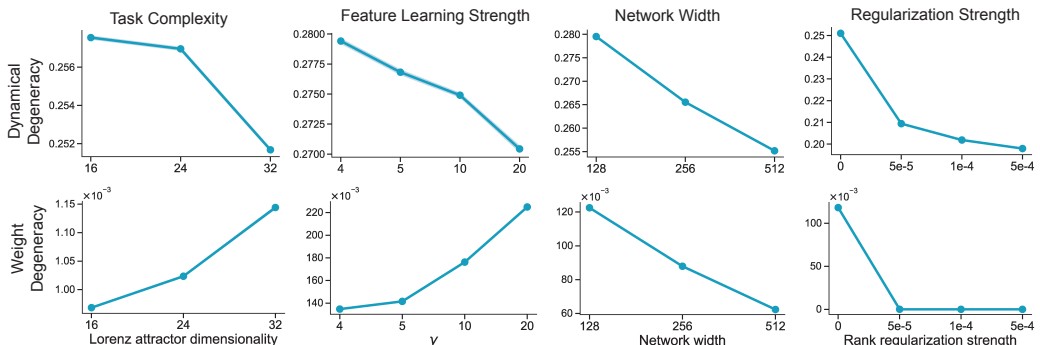

Figure 11: Varying the four factors on Lorenz 96 next-step prediction task changes solution degeneracy across the dynamical and weight level in a way that is consistent with Table 1.

# F   Additional details on the degeneracy metrics

## F.1   Dynamical Degeneracy

Briefly, DSA proceeds as follows: Given two RNNs with hidden states $\mathbf{h}_1(t) \in \mathbb{R}^n$ and $\mathbf{h}_2(t) \in \mathbb{R}^n$, we first generate a delay-embedded matrix, $\mathbf{H}_1$ and $\mathbf{H}_2$ of the hidden states in their original state space. Next, for each delay-embedded matrix, we use Dynamic Mode Decomposition (DMD) [96] to extract linear forward operators $\mathbf{A}_1$ and $\mathbf{A}_2$ of the two systems' dynamics. Finally, a Procrustes distance between the two matrices $\mathbf{A}_1$ and $\mathbf{A}_2$ is used to quantify the dissimilarity between the two dynamical systems and provide an overall DSA score, defined as:

$$d_{\text{Procrustes}}(\mathbf{A}_1, \mathbf{A}_2) = \min_{\mathbf{Q} \in O(n)} \|\mathbf{A}_1 - \mathbf{Q}\mathbf{A}_2\mathbf{Q}^{-1}\|_F$$

where $\mathbf{Q}$ is a rotation matrix from the orthogonal group $O(n)$ and $\|\cdot\|_F$ is the Frobenius norm. This metric quantifies how dissimilar the dynamics of the two RNNs are after accounting for orthogonal transformations. We quantify Dynamical Degeneracy across many RNNs as the average pairwise distance between pairs of RNN neural-dynamics (hidden-state trajectories).

After training, we extract each network's hidden-state activations for every trial in the training set, yielding a tensor of shape (trials × time steps × neurons). We collapse the first two dimensions and yield a matrix of size (trials × time steps) × neurons. We then apply PCA to retain the components that explain 99% of the variance to remove noisy and low-variance dimensions of the hidden state trajectories. Next, we perform a grid search over candidate delay lags, with a minimum lag of 1 and a maximum lag of 30, selecting the lag that minimizes the reconstruction error of DSA on the dimensionality reduced trajectories. Finally, we fit DSA with full rank and the optimal lag to these PCA-projected trajectories and compute the pairwise DSA distances between all networks.

## F.2   Weight degeneracy

We computed the pairwise distance between the recurrent matrices from different networks using Two-sided Permutation with One Transformation [97, 98] function from the Procrustes Python package [99].

## F.3   Establishing a null distribution for dynamical and weight degeneracy

The DSA scores that we used to define the dynamical degeneracy are inherently context-dependent. Specifically, the absolute scale of DSA distances can vary with hyperparameters, particularly the delay embedding dimension and the rank used in DSA, because the underlying Procrustes analysis between two dynamics matrices relies on the Frobenius norm, which in turn depends on the dimension of the dynamic operator being compared. Following the procedure described in the original DSA paper, we fixed these hyperparameters across all groups within each task to ensure fair comparison.

To further validate the interpretation of DSA values, we computed null distributions of the DSA scores, i.e. the distribution of DSA scores when sampled neural activities come from *identical* networks. For

each of the 50 networks analyzed in Figure 3B, we randomly split the sampled neural trajectories from the same network into two subsets and computed DSA distances between them. This procedure yields a distribution of DSA scores expected from *identical dynamical systems*, which serves as a reference noise floor. The 95% confidence intervals (CIs) for these null distributions are reported below (header labels such as "Level 1" correspond to the task-complexity levels shown in the main-text figures). These CIs are, on average, an order of magnitude smaller than the computed dynamical degeneracy, indicating that the observed differences between networks trained from different initializations are statistically significant.

Table 7: Establishing a null distribution for dynamical degeneracy: 95% confidence intervals of null DSA scores computed by comparing trajectories from the same network. CIs are on average an order of magnitude smaller than across-network distances.

| Task Complexity | Level 1 | Level 2 | Level 3 | Level 4 |
|---|---|---|---|---|
| N-BFF | [0.011, 0.013] | [0.009, 0.016] | [0.008, 0.013] | [0.006, 0.009] |
| Delayed Discrimination | [0.039, 0.064] | [0.014, 0.076] | [0.025, 0.032] | — |
| Sine Wave Generation | [0.057, 0.102] | [0.054, 0.081] | [0.048, 0.073] | — |
| Path Integration | [0.023, 0.037] | [0.010, 0.018] | — | — |

For the PIF distance we used to define weight degeneracy, we similarly established a noise floor by *randomly permuting* each trained network's recurrent weight matrix and computing the distance between the permuted and original matrices. The PIF metric reliably recovers a PIF distance of 0 under this null setting, confirming its robustness to noise and the meaningfulness of the reported cross-network PIF differences.

# G   Representational degeneracy

We further quantified solution degeneracy at the representational level—that is, the variability in each network's internal feature space when presented with the same input dataset—using Singular Vector Canonical Correlation Analysis (SVCCA). SVCCA works by first applying singular value decomposition (SVD) to each network's activation matrix, isolating the principal components that capture most of its variance, and then performing canonical correlation analysis (CCA) to find the maximally correlated directions between the two reduced subspaces. The resulting canonical correlations therefore measure how similarly two networks represent the same inputs: high average correlations imply low representational degeneracy (i.e., shared feature subspaces), whereas lower correlations reveal greater divergence in what the models learn. We define the representational degeneracy (labeled as the SVCCA distance below) as

$$d_{\mathrm{repr}}(A_x, A_y) \;=\; 1 \;-\; \mathrm{SVCCA}\big(A_x, A_y\big).$$

We found that as we vary the four factors that robustly control the dynamical degeneracy across task-trained RNNs, the representational-level degeneracy isn't necessarily constrained by those same factors in the same way. In RNNs, task-relevant computations are implemented at the level of network's dynamics instead of static representations, and RNNs that implement similar temporal dynamics can have disparate representaional geometry. Therefore, it is expected that task complexity, learning regime, and network size change the task-relevant computations learned by the networks by affecting their neural dynamics instead of representations. DSA captures the dynamical aspect of the neural computation by fitting a forward operator matrix $A$ that maps the network's activity at one time step to the next, therefore directly capturing the temporal evolution of neural activities. By contrast, SVCCA aligns the principal subspaces of activation vectors at each time point but treats those vectors as independent samples—it never examines how one state evolves into the next. As a result, SVCCA measures only static representational similarity and cannot account for the temporal dependencies that underlie RNN computations. Nonetheless, we expect SVCCA might be more helpful in measuring the solution degeneracy in feedforward networks.

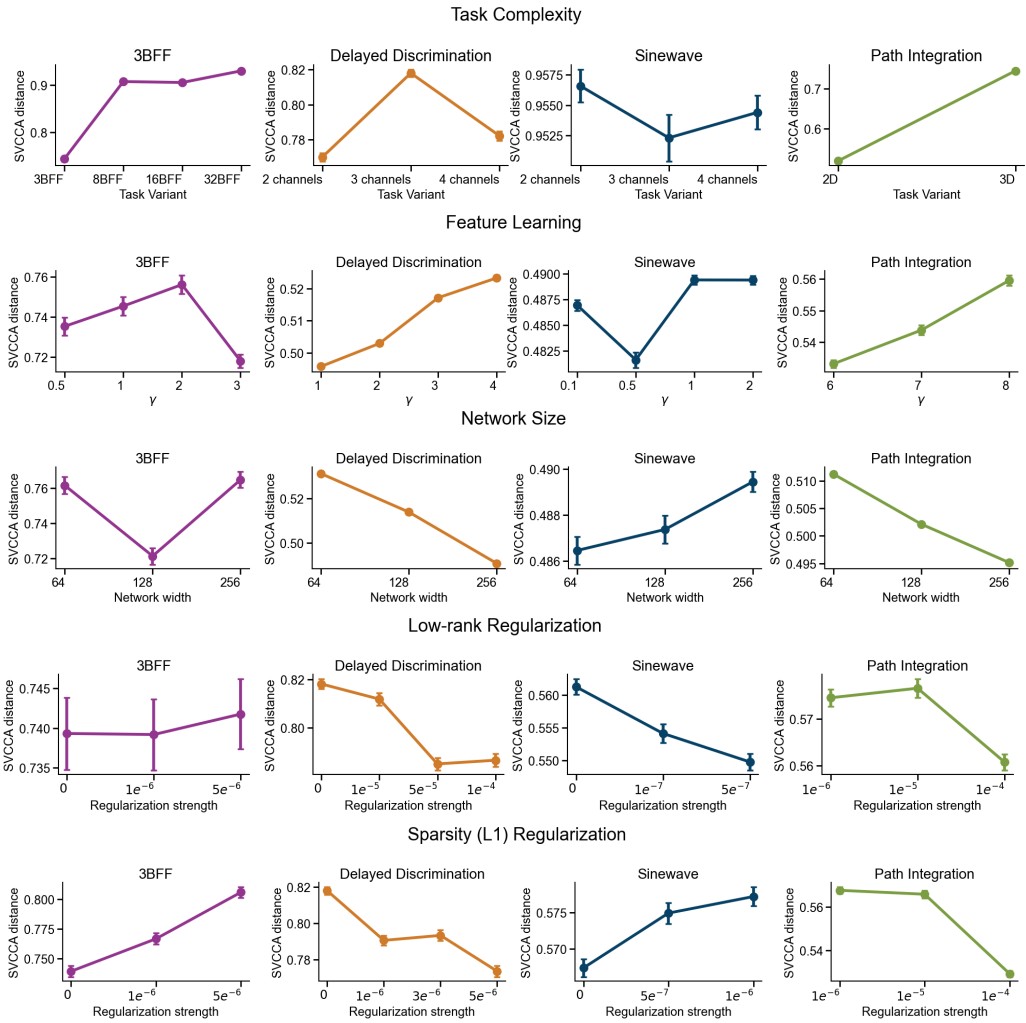

Figure 12: Representational degeneracy, as measured by the average SVCCA distance between networks, does not necessarily change uniformly as we vary task complexity, feature learning strength, network size, and regularization strength.

# H  Task complexity effect on degeneracy in Gated RNNs

To examine whether the observed trends in dynamical and weight degeneracy generalize beyond vanilla RNNs, we conducted additional experiments using gated recurrent units (GRUs). Note that prior work suggests that architectural choices influence the *geometry* but not the *topology* of neural dynamics, which is primarily shaped by task structure [100]. Meanwhile, the Dynamical Similarity Analysis (DSA) metric we employ to quantify dynamical degeneracy is designed to precisely capture the topological organization of neural dynamics while remaining invariant to geometric transformations [40].

As a preliminary test, we trained GRUs on the Sine Wave Generation task while systematically varying task complexity by changing the number of input–output channels. Consistent with our findings in vanilla RNNs, increasing task complexity led to a **decrease in dynamical degeneracy** and a **rise in weight degeneracy**.

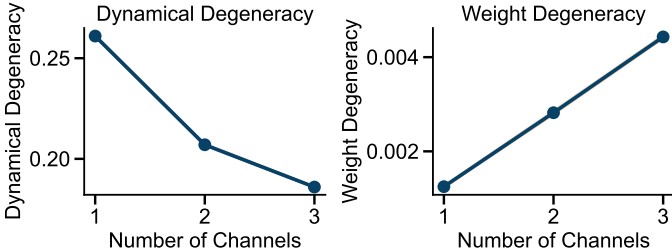

Figure 13: Increasing task complexity in the Sine Wave Generation task produces the same effect on dynamical and weight degeneracy in both vanilla RNNs and GRUs.

# I  Dense sweep on feature learning, network width, and regularization strength

it is important to know whether the degeneracy trends generalize to intermediate values and beyond the ranges reported in the main paper. To test this, we used 3-BFF as an example and ran a dense sweep both interpolating within and extrapolating beyond the ranges shown in Figs. 3 and 6–8. We demonstrate that cross dynamical and weight levels, the degeneracy trends remain consistent and interpolate smoothly across these intermediate values.

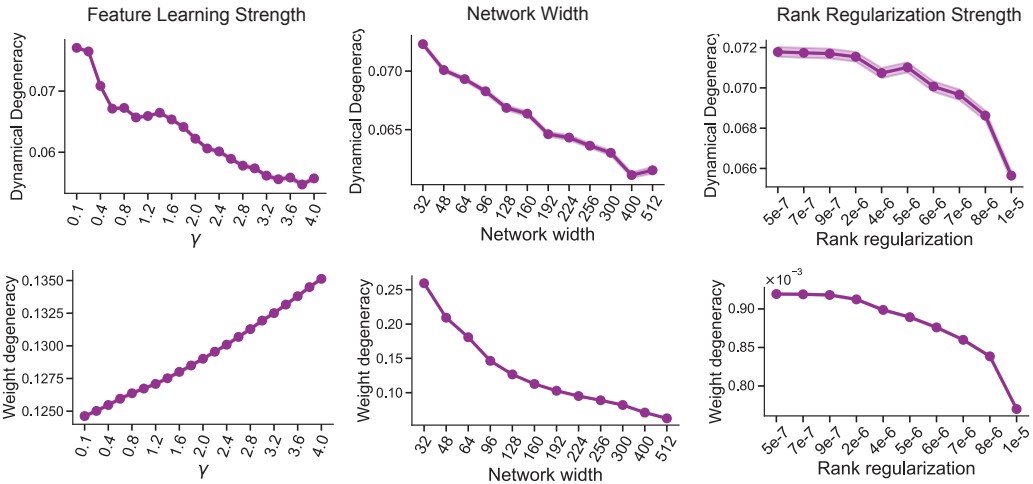

Figure 14: Feature learning, network width, and regularization strength's effect on degeneracy over a denser sweep of conditions on the 3-Bits Flip Flop task.

# J   Detailed characterization of OOD generalization performance

In addition to showing the behavioral degeneracy in the main text, here we provide a more detailed characterization of the OOD behavior of networks by showing the mean versus standard deviation, and the distribution of the OOD losses.

## J.1   Changing task complexity

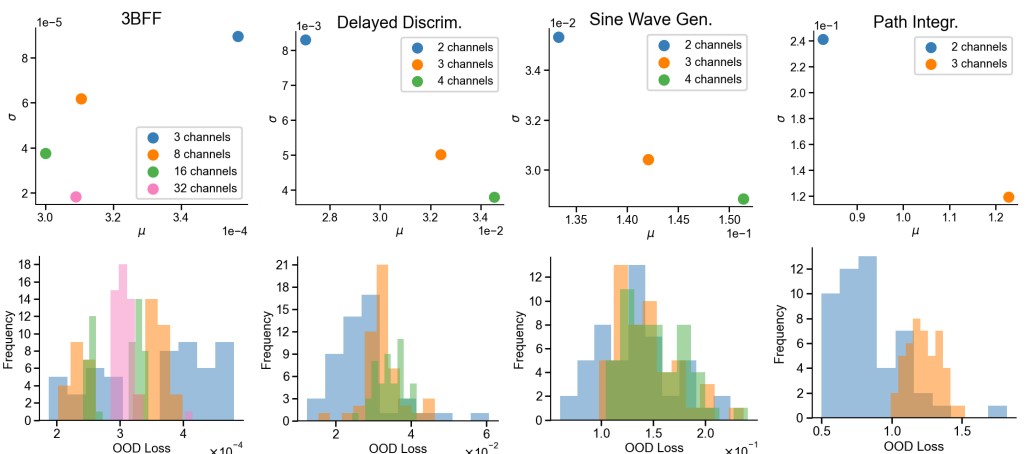

Figure 15: Detailed characterization of the OOD performance of networks while changing task complexity.

## J.2   Changing feature learning strength

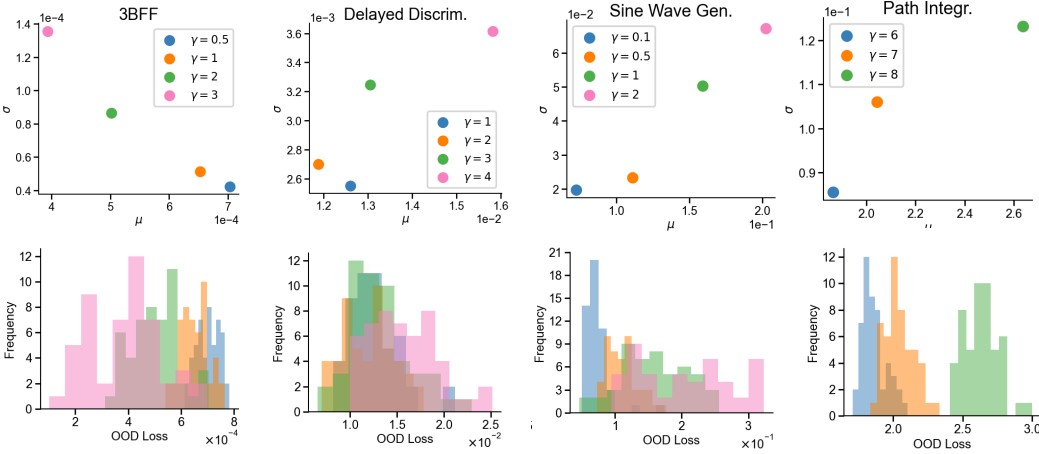

Figure 16: Detailed characterization of the OOD performance of networks while changing feature learning strength. Across Delayed Discrimination, Sine Wave Generation, and Path Integration tasks, networks trained with larger $\gamma$ – and thus undergoing stronger feature learning – exhibit higher mean OOD generalization loss together with higher variability, potentially reflecting overfitting to the training task.

## J.3 Changing network size

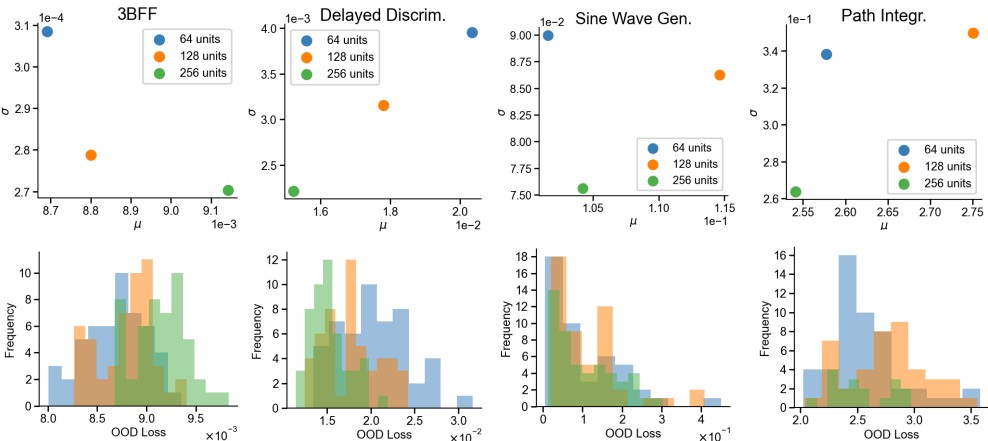

Figure 17: Detailed characterization of the OOD performance of networks while changing network size.

## J.4 Changing regularization strength

### J.4.1 Low-rank regularization

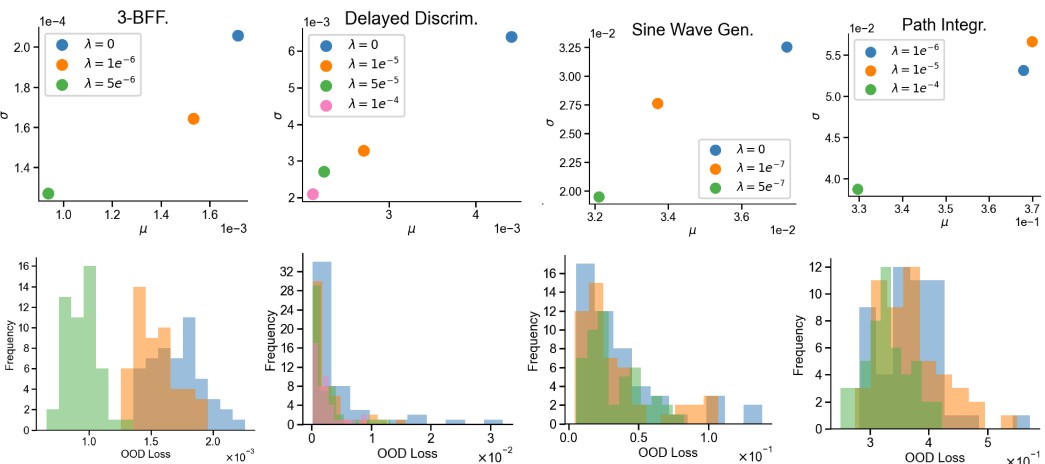

Figure 18: Detailed characterization of the OOD performance of networks while changing low-rank regularization strength.

### J.4.2 Sparsity (L1) regularization

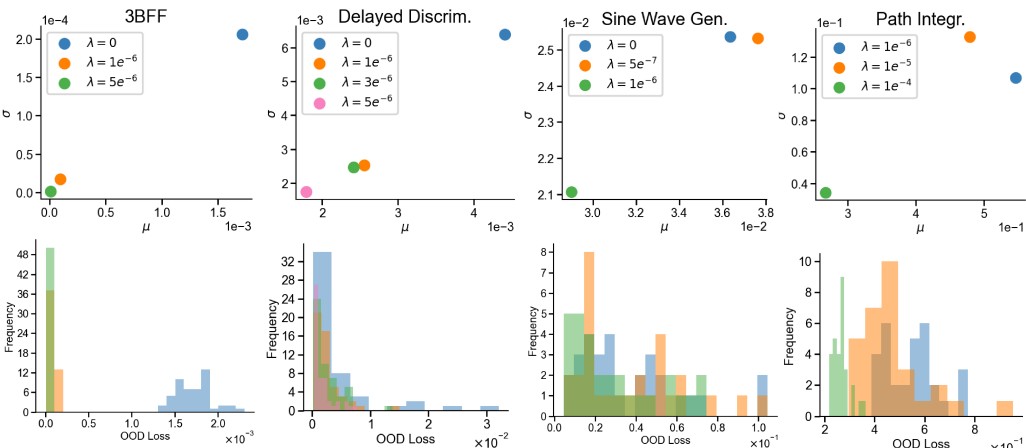

Figure 19: Detailed characterization of the OOD performance of networks while changing sparsity (L1) regularization strength.

## K  A short introduction to Maximal Update Parameterization ($\mu P$)

Under the NTK parametrization, as the network width goes to infinity, the network operates in the *lazy* regime, where its functional evolution is well-approximated by a first-order Taylor expansion around the initial parameters [73**?**, 54, 55]. In this limit feature learning is suppressed and training dynamics are governed by the fixed Neural Tangent Kernel (NTK).

To preserve non-trivial feature learning at large width, the *Maximal Update Parametrization* ($\mu P$) rescales both the weight initialisation and the learning rate. $\mu P$ keeps three quantities *width-invariant* at every layer—(i) the norm/variance of activations (ii) the norm/variance of the gradients, and (iii) the parameter updates applied by the optimizer [101, 102, 56, 57].

For recurrent neural networks, under Stochastic Gradient Descent (SGD), the network output, initialization, and learning rates are scaled as

$$f = \frac{1}{\gamma_0 N}\, \vec{w} \cdot \phi(h), \tag{1}$$

$$\partial_t h = -h + \frac{1}{\sqrt{N}}\, J\,\phi(h), \qquad J_{ij} \sim \mathcal{N}(0,1), \tag{2}$$

$$\eta_{\text{SGD}} = \eta_0\,\gamma_0^2\,N. \tag{3}$$

Under Adam optimizer, the network output, initialization, and learning rates are scaled as

$$f = \frac{1}{\gamma_0 N}\, \vec{w} \cdot \phi(h), \tag{4}$$

$$\partial_t h = -h + \frac{1}{N}\, J\,\phi(h), \qquad J_{ij} \sim \mathcal{N}(0,N), \tag{5}$$

$$\eta_{\text{Adam}} = \eta_0\,\gamma_0. \tag{6}$$

## L  Theoretical relationship between parameterizations

We compare two RNN formalisms used in different parts of the main manuscript: a standard discrete-time RNN trained with fixed learning rate and conventional initialization, and a $\mu P$-style RNN trained with leaky integrator dynamics and width-aware scaling.

In the standard discrete-time RNN, the hidden activations are updated as

$$h(t+1) = \phi\big(W_h h(t) + W_x x(t)\big),$$

In $\mu P$ RNNs, the hidden activations are updated as

$$h(t+1) - h(t) = \tau\left(-h(t) + \frac{1}{N}J\phi(h(t)) + Ux(t)\right)$$

When $\tau = 1$,

$$h(t+1) - h(t) = -h(t) + \frac{1}{N}J\phi(h(t)) + Ux(t)$$

$$h(t+1) = \frac{1}{N}J\phi(h(t)) + Ux(t)$$

Aside from the overall scaling factor, the difference between the two parameterizations lies in the placement of the non-linearity:

- **Standard RNN:** $\phi$ is applied *post-activation*, i.e. after the recurrent and input terms are linearly combined,
- **$\mu$P RNN:** $\phi$ is applied *pre-activation*; i.e. before the recurrent weight matrix, so the hidden state is first non-linearized and then linearly combined

Miller and Fumarola [103] demonstrated that two classes of continuous-time firing-rate models which differ in their placement of the non-linearity are mathematically equivalent under a change of variables:

$$\text{v-model} \quad \tau\frac{dv}{dt} = -v + \tilde{I}(t) + Wf(v)$$

$$\text{r-model:} \quad \tau\frac{dr}{dt} = -r + f(Wr + I(t))$$

with equivalence holding under the transformation $v(t) = Wr(t) + I(t)$ and $\tilde{I}(t) = I(t) + \tau\frac{dI}{dt}$, assuming matched initial conditions.

Briefly, they show that $Wr + I$ evolves according to the $v$-equation as follows:

$$v(t) = Wr(t) + I(t)$$
$$\frac{dv}{dt} = \frac{d}{dt}\left(Wr(t) + I(t)\right)$$
$$= W\frac{dr}{dt} + \frac{dI}{dt}$$
$$= W\left(\frac{1}{\tau}\left(-r + f(Wr + I)\right)\right) + \frac{dI}{dt}$$
$$\tau\frac{dv}{dt} = -Wr + Wf(Wr + I) + \tau\frac{dI}{dt}$$
$$= -(v - I) + Wf(v) + \tau\frac{dI}{dt}$$
$$= -v + I + \tau\frac{dI}{dt} + Wf(v)$$
$$\tau\frac{dv}{dt} = -v + \tilde{I}(t) + Wf(v)$$

This mapping applies directly to RNNs viewed as continuous-time dynamical systems and helps relate v-type $\mu$P-style RNNs to standard discrete-time RNNs. It suggests that the $\mu$P RNN (in v-type form) and the standard RNN (in r-type form) can be treated as different parameterizations of the same underlying dynamical system when:

- Initialization scales are matched
- The learning rate is scaled appropriately with $\gamma$

- Output weight norms are adjusted according to width

In summary, while a theoretical equivalence exists, it is contingent on consistent scaling across all components of the model. In this manuscript, we use the standard discrete-time RNNs due to its practical relevance for task-driven modeling community, while switching to $\mu P$ to isolate the effect of feature learning and network size. Additionally, we confirm that the feature learning and network size effects on degeneracy hold qualitatively the same in standard discrete-time RNNs, unless where altering network width induces unstable and lazier learning in larger networks (Figure P and Q).

## M  Verifying larger $\gamma$ reliably induces stronger feature learning in $\mu P$

In $\mu P$ parameterization, the parameter $\gamma$ interpolates between lazy training and rich, feature-learning dynamics, without itself being the absolute magnitude of feature learning. Here, we assess feature-learning strength in RNNs under varying $\gamma$ using two complementary metrics:

**Weight-change norm** which measures the magnitude of weight change throughout training. A larger weight change norm indicates that the network undergoes richer learning or more feature learning.

$$\frac{\|\mathbf{W}_T - \mathbf{W}_0\|_F}{N},$$

where N is the number of parameters in the weight matrices being compared.

**Kernel alignment (KA)**, which measures the directional change of the neural tangent kernel (NTK) before and after training. A lower KA score corresponds to a larger NTK rotation and thus stronger feature learning.

$$\text{KA}\big(K^{(f)}, K^{(0)}\big) \;=\; \frac{\text{Tr}\big(K^{(f)} K^{(0)}\big)}{\big\|K^{(f)}\big\|_F \big\|K^{(0)}\big\|_F}, \qquad K \;=\; \nabla_W \hat{y}^\top \nabla_W \hat{y}.$$

We demonstrate that higher $\gamma$ indeed amplifies feature learning inside the network.

### M.1  N-BFF

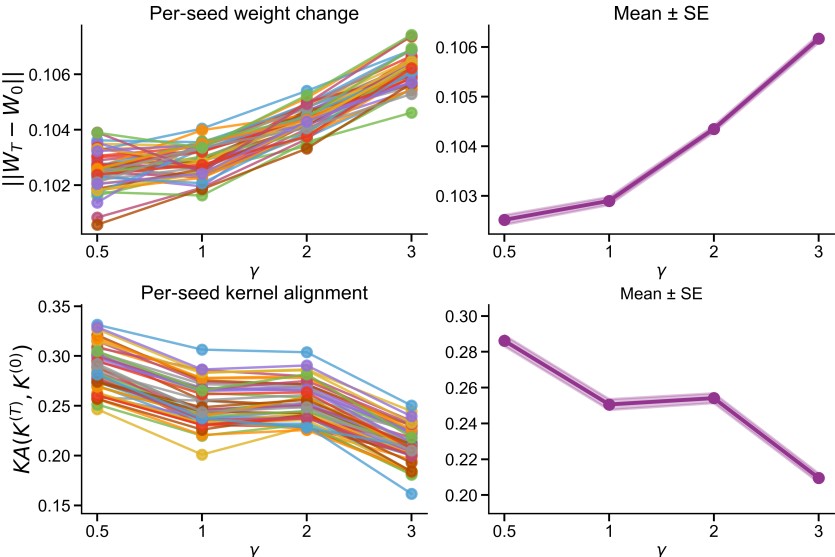

Figure 20: Weight change norm and kernel alignment for networks trained on the 3-Bits Flip Flop task as we vary $\gamma$. On the left panels, we show the per-seed metrics where connected dots of the same color are networks of identical initialization trained with different $\gamma$. On the right panels, we show the mean and standard error of the metrics across 50 networks. For larger $\gamma$, the weights move further from their initializations as shown by the larger weight change norm, and their NTK evolves more distinct from the network's NTK at initialization as shown by the reduced KA. Both indicate stronger feature learning for networks trained under larger $\gamma$.

## M.2 Delayed Discrimination

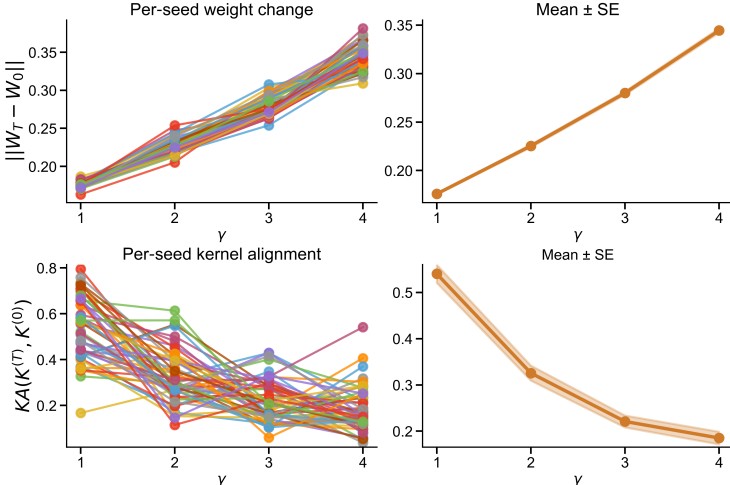

Figure 21: Stronger feature learning for networks trained under larger $\gamma$ on the Delayed Discrimination task.

## M.3 Sine Wave Generation

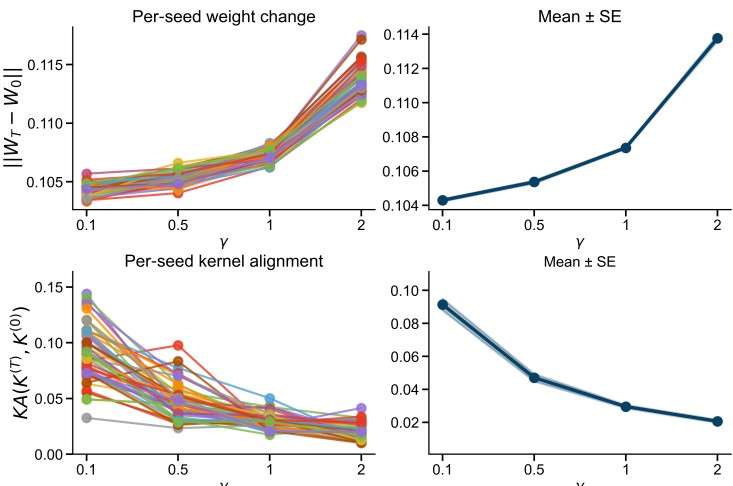

Figure 22: Stronger feature learning for networks trained under larger $\gamma$ on the Sine Wave Generation task.

## M.4  Path Integration

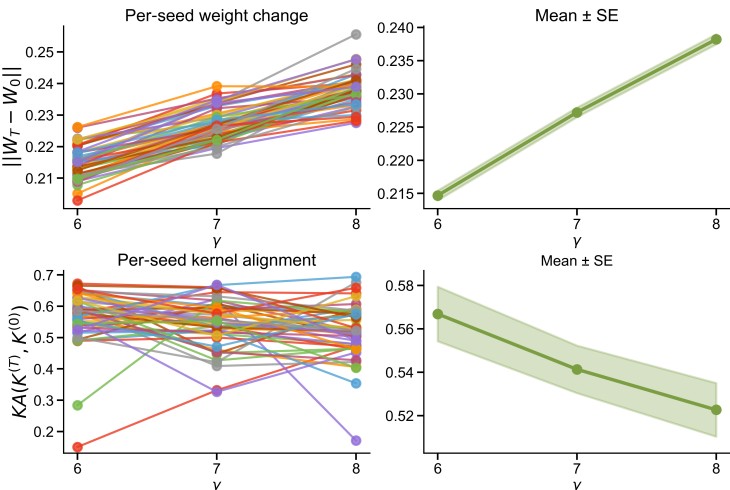

Figure 23: Stronger feature learning for networks trained under larger $\gamma$ on the Path Integration task.

# N  Verifying $\mu P$ reliably controls for feature learning across network width

Here, we only use Kernel Alignment to assess the feature learning strength in the networks since the unnormalized weight-change norm $\|\mathbf{W}_T - \mathbf{W}_0\|_F$ scales directly with matrix size (therefore network size) and there exists no obvious way to normalize across different dimensions. In our earlier analysis where we compared weight-change norms at varying $\gamma$, network size remained fixed, so those Frobenius-norm measures were directly comparable. We found that, for all tasks except Delayed Discrimination, the change in mean KA across different network sizes remains extremely small (less than 0.1), which demonstrates that $\mu P$ parameterization with the same $\gamma$ has effectively controlled for feature learning strength across network sizes. On Delayed Discrimination, the networks undergo slightly lazier learning for larger network sizes. Nevertheless, we still include Delayed Discrimination in our analyses of solution degeneracy to ensure *our conclusions remain robust even when $\mu P$ can't perfectly equalize feature-learning strength across widths.* As shown in the main paper, lazier learning regime generally increases dynamical degeneracy; yet, larger networks which exhibit lazier learning in the N-BFF task actually display lower dynamical degeneracy. This reversed trend confirms that the changes in solution degeneracy arise from network size itself, not from residual variation in feature learning strength.

## N.1  N-BFF

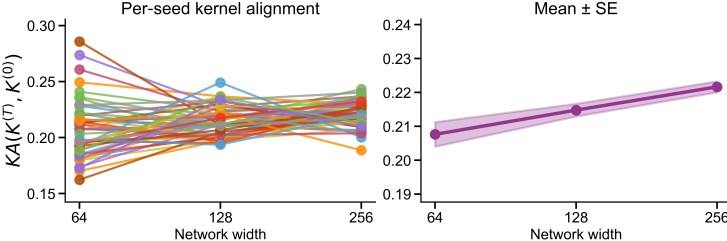

Figure 24: Kernel alignment (KA) for different network width on the 3 Bits Flip-Flop task. (Lower KA implies more feature learning.)

## N.2 Delayed Discrimination

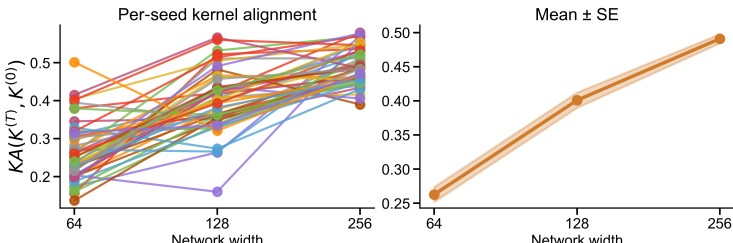

Figure 25: Kernel alignment for different network width on the Delayed Discrimination task.

## N.3 Sine Wave Generation

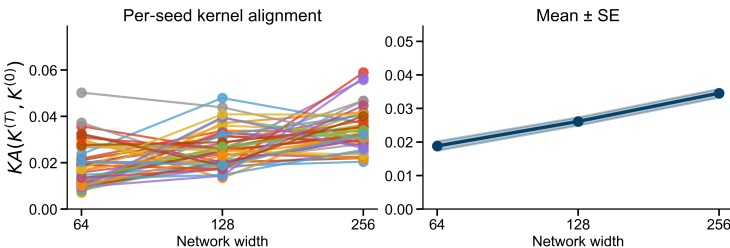

Figure 26: Kernel alignment for different network width on the Sine Wave Generation task.

## N.4 Path Integration

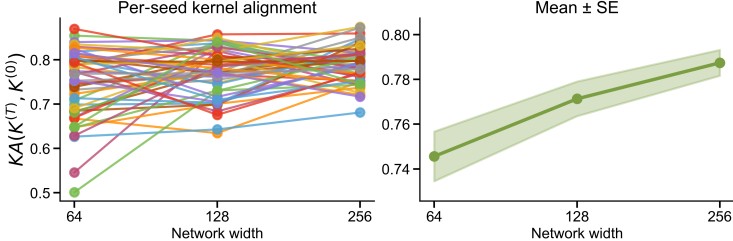

Figure 27: Kernel alignment for different network width on the Path Integration task.

# O Regularization's effect on degeneracy for all tasks

In addition to showing regularization's effect on degeneracy in Delayed Discrimination task in the main paper, here we show that heavier low-rank regularization and sparsity regularization also reliably reduce solution degeneracy across neural dynamics, weights, and OOD behavior in the other three tasks.

## O.1 Low-rank regularization

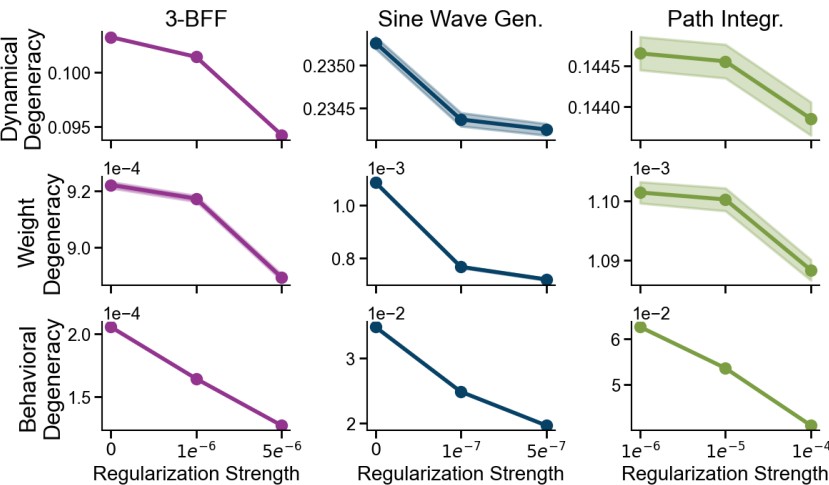

Figure 28: Low-rank regularization reduces degeneracy across neural dynamics, weight, and OOD behavior on the N-BFF, Sinewave Generation, and Path Integration task.

## O.2 Sparsity regularization

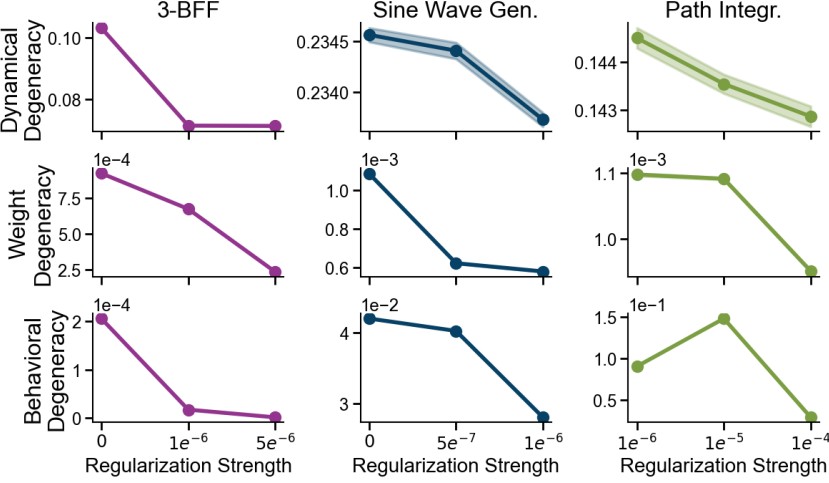

Figure 29: Sparsity regularization reduces degeneracy across neural dynamics, weight, and OOD behavior on the N-BFF, Sinewave Generation, and Path Integration task.

# P   Test feature learning effect on degeneracy in standard parameterization

While $\mu P$ lets us systematically vary feature-learning strength to study its impact on solution degeneracy, we confirm that the same qualitative pattern appears in *standard* discrete-time RNNs: stronger feature learning **lowers dynamical degeneracy** and **raises weight degeneracy** (Figure 30).

To manipulate feature-learning strength in these ordinary RNNs we applied the $\gamma$-**trick**—scaling the network's outputs by $\gamma$—and multiplied the learning rate by the same factor. With width fixed, these two operations replicate the effective changes induced by $\mu P$. Figure 31 shows that this combination reliably tunes feature-learning strength. Besides weight-change norm and kernel alignment, we also report **representation alignment (RA)**, giving a more fine-grained view of how much the learned features deviate from their initialization [62]. Representation alignment is the directional change of the network's representational dissimilarity matrix before and after training, and is defined by

$$\mathrm{RA}\big(R^{(T)}, R^{(0)}\big) := \frac{\mathrm{Tr}\big(R^{(T)} R^{(0)}\big)}{\|R^{(T)}\| \, \|R^{(0)}\|}, \qquad R := H^\top H,$$

A lower RA means more change in the network's representation of inputs before and after training, and indicates stronger feature learning.

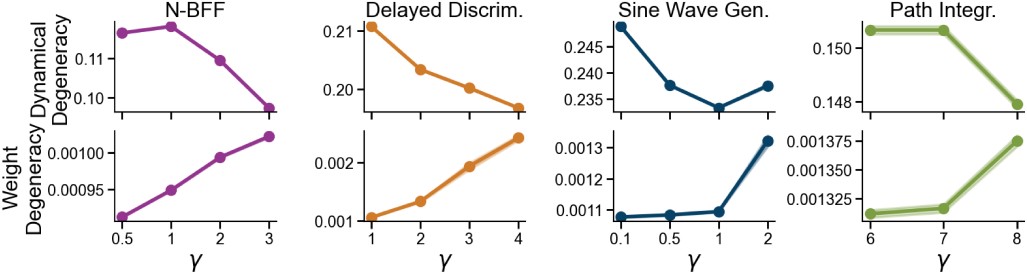

Figure 30: Stronger feature learning reliably decreases dynamical degeneracy while increasing weight degeneracy in standard discrete-time RNNs.

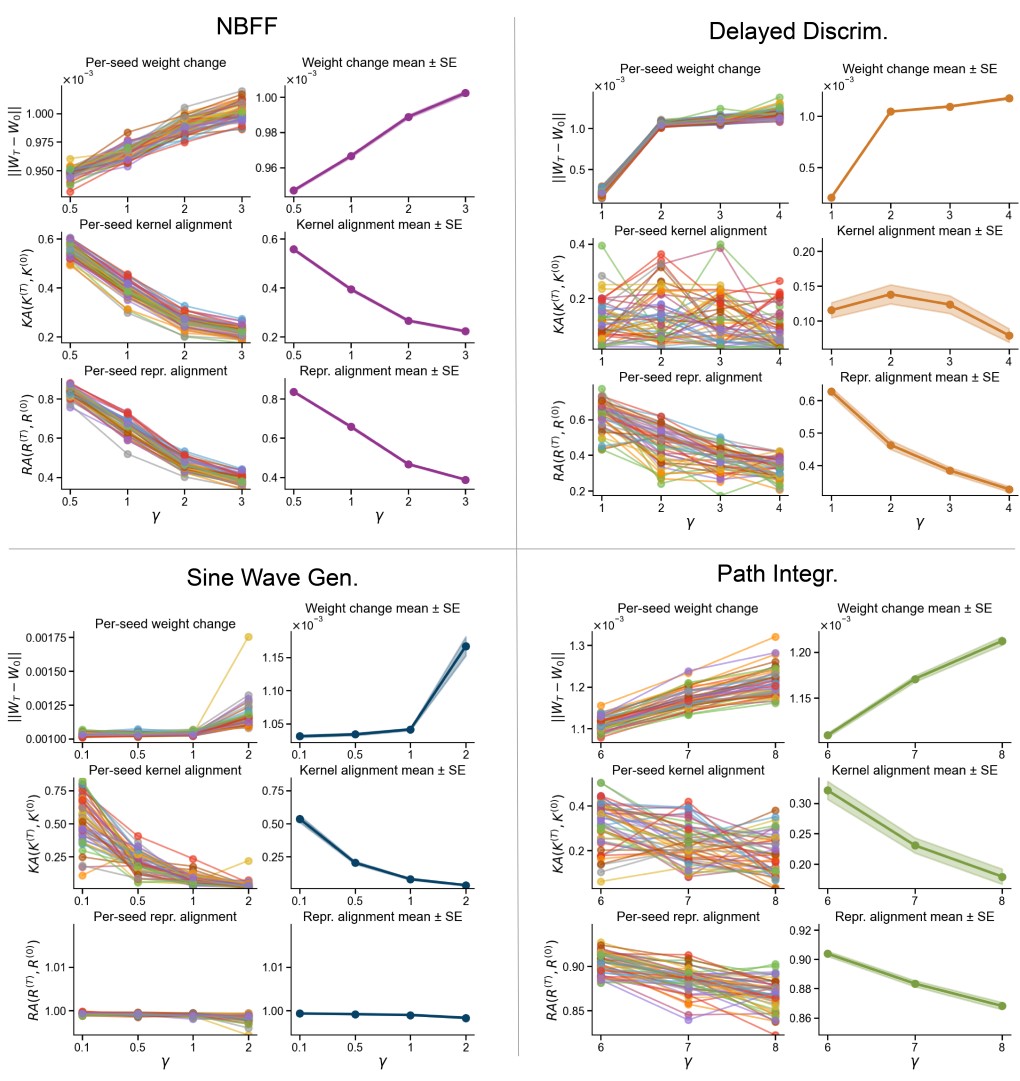

Figure 31: Larger $\gamma$ reliably induces stronger feature learning in standard discrete-time RNNs.

# Q    Test network size effect on degeneracy in standard parameterization

When we vary network width, both the standard parameterization and $\mu P$ parameterization display the same overall pattern: **larger networks exhibit lower dynamical and weight degeneracy**. An exception arises in the 3BFF task, where feature learning becomes unstable and collapses in the wider models. In that setting we instead see *higher* dynamical degeneracy, which we suspect because the feature learning effect (lazier learning leads to higher dynamical degeneracy) dominates the network size effect.

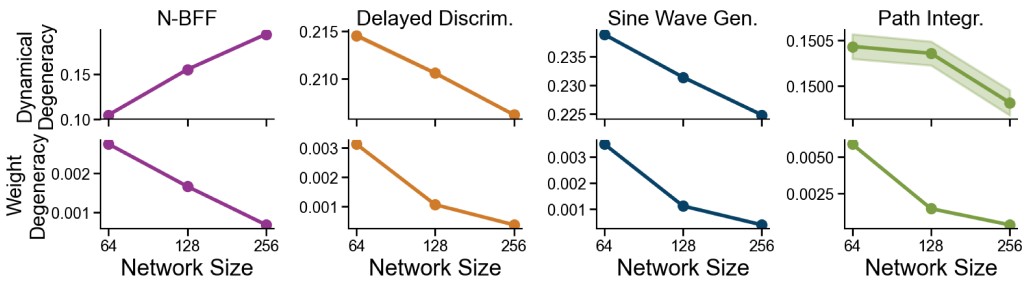

Figure 32: Larger network sizes lead to lower dynamical and weight degeneracy, except in the case where feature learning is unstable across width (in N-BFF).

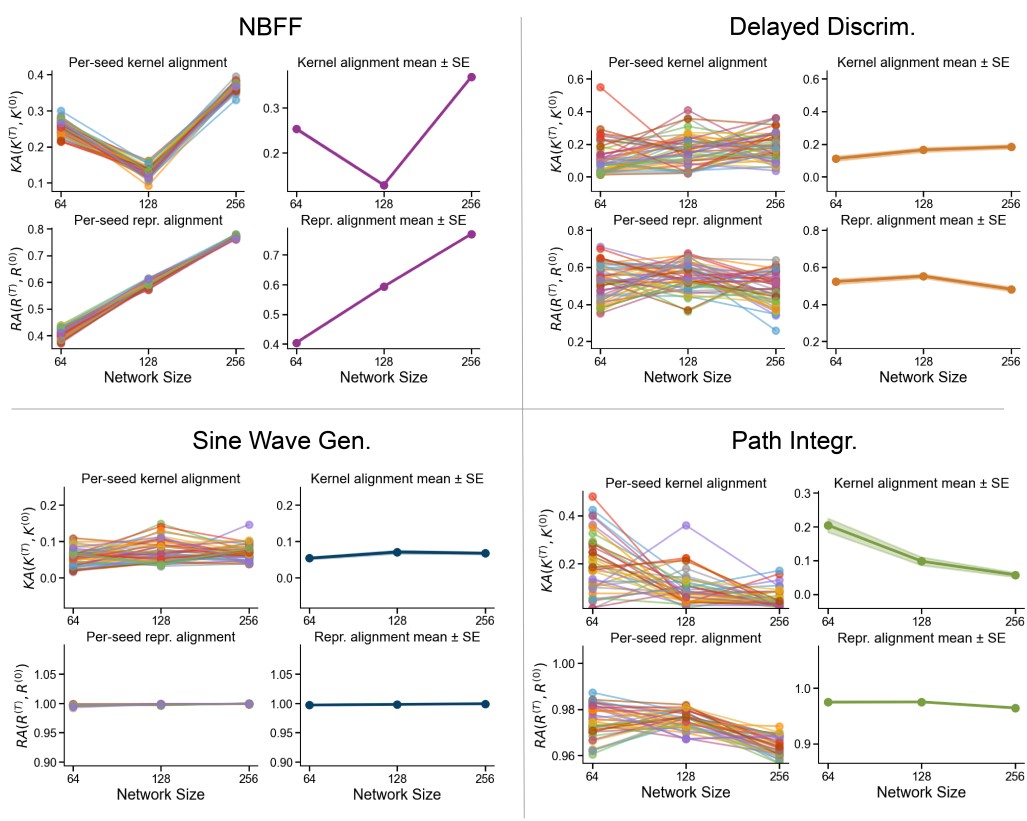

Figure 33: When changing network width in standard discrete-time RNNs, feature learning strength remains stable across width except in N-BFF, where notably lazier learning happens in the widest network.

# R  Disclosure of compute resources

In this study, we conducted 50 independent training runs on each of four tasks, systematically sweeping four factors that modulate solution degeneracy—task complexity (15 experiments), learning regime (15 experiments), network size (12 experiments), and regularization strength (26 experiments), resulting in a total of 3400 networks. Each experiment was allocated 5 NVIDIA V100/A100 GPUs, 32 CPU cores, 256 GB of RAM, and a 4-hour wall-clock limit, for a total compute cost of approximately 68 000 GPU-hours.

