# OpenReview forum: "Measuring and Controlling Solution Degeneracy across Task-Trained Recurrent Neural Networks"
_NeurIPS.cc/2025/Conference — NeurIPS 2025 spotlight_

### Official Review · Reviewer_fDhN · 2025-06-30

**Clarity:** 3
**Significance:** 3
**Originality:** 2
**Rating:** 5
**Confidence:** 3

**Summary:**

This paper investigates the solution degeneracy of recurrent neural networks (RNNs) trained on a suite of tasks, evaluating solution diversity across behavioral, dynamical, and weight-space levels. The central claim is that task complexity predictably modulates degeneracy: increasing complexity leads to more similar neural dynamics across training runs, but greater divergence in weight space. The authors use recently proposed tools such as Dynamical Similarity Analysis (DSA) and introduce the Permutation-Invariant Frobenius distance (PIF) to quantify similarity between trained RNNs. The paper proposes methods to quantify and control degeneracy (e.g., via task complexity, auxiliary loss functions, regularization).

**Questions:**

•	The paper needs stronger empirical or theoretical validation for DSA and PIF as measures of degeneracy. What threshold should count as significant difference in DSA or PIF? Why is a DSA of ~0.12 meaningful if previous work considered that range as "invariant"?

•	Can the authors provide flow fields, phase portraits, or other dynamical visualizations to show the difference in dynamics?

•	How do the measures and trends extend to chaotic or high-dimensional RNNs? For instance, are the same trends visible with chaotic attractors?

•	In Figure 3, are network sizes held constant across varying complexities? Could increasing complexity simply lead to networks using more of their capacity, thereby reducing variability?

•	The paper does not report task performance. How well do these RNNs actually solve the tasks? Are higher-degeneracy networks lower-performing?

•	The paper refers to dispersion of local minima but does not show empirical support. Could the observed weight variability be due to wide basins in a large solution space?

•	Are DSA comparisons based on autonomous (free-running) dynamics or input-driven trajectories? This distinction is crucial for interpreting the meaning of “similar dynamics.”

**Ethical Concerns:**

["NO or VERY MINOR ethics concerns only"]

**Final Justification:**

All of my concerns have been addressed by the authors, and they have provided additional results that directly respond to the major issues I raised.
These new results substantially strengthen the paper.
Provided that the described changes and additions are incorporated into the final version, I recommend accepting the paper.
Based on the other reviews, it appears that all (or most) remaining concerns have also been addressed.

**Limitations:**

While the idea of the paper is interesting and novel, some limitations remain:

•	Lack of analysis for chaotic or more realistic dynamical systems.

•	Absence of flow field visualizations to confirm qualitative differences in neural dynamics.

•	Lack of explicit model performance reporting.

•	Limited number of datapoints (e.g., 2–4 variants per task) which weakens broad generalization claims.

**Paper Formatting Concerns:**

•	There is a small typo in the section title of 3.1

**Quality:**

2

**Strengths And Weaknesses:**

Strengths:

•	The paper is clearly written, with well-organized figures and a clean narrative structure

•	The focus on degeneracy and its controllability is relevant for both interpretability and robustness in ML models

Weaknesses:


•	The validation of the proposed complexity and degeneracy measures is limited. For example, DSA scores in the paper in part (e.g. Fig. 3B and Fig. 6) show values on the order of 0.12—similar to what's considered "invariant" in previous DSA work (e.g., Ostrow et al. 2023, Fig. 4,[1]). Without a stronger argument for statistical significance, these differences appear too small to support strong claims

•	Several key claims (e.g., the significance of degeneracy differences, the effect of “dispersion of local minima”) are made without sufficient visual or quantitative evidence. The paper would benefit from more direct visualizations of flow fields or attractors, especially for chaotic or high-dimensional tasks, to support the dynamical claims

•	The scale of experiments is modest—typically 2–4 data points per task variant—which limits the strength of the generalization claims (e.g. Figs. 3,6,7). Tasks are also relatively low-dimensional and may not reflect the complexity of real-world RNN applications




References
[1] Ostrow, M., Eisen, A., Kozachkov, L., & Fiete, I. (2023). Beyond geometry: Comparing the temporal structure of computation in neural circuits with dynamical similarity analysis. Advances in Neural Information Processing Systems, 36, 33824-33837.

---

> ### Author Rebuttal · Authors · 2025-07-30
>
> Thank you for your thoughtful and constructive questions! We are glad you find this work well-written and relevant for both interpretability and robustness in ML models research.
>
> > Q1: What counts as a meaningful difference in DSA and PIF?
>
> * **DSA scores are context dependent**.
> We would like to point out that DSA does not provide a context-invariant score but rather the scale of DSA distance can depend on hyperparameters (particularly the delay and rank of DSA) since the Procrustes analysis over two dynamics matrices which relies on the *Frobenius* norm is not scale-invariant.
> To select these hyperparameters, we followed the original DSA paper's process. We carefully keep these hyperparameters the same for all groups within each task for a fair comparison. In figures 3,4,6-8, we report the average pairwise DSA distance with std. error bands that clearly indicate when mean pairwise DSA differences between groups are significant.
>
> * **Calculating null distributions for DSA scores**:  For each of our 50 networks (in Fig 3B), we split the sampled neural trajectories from each network into 2 subsets, and computed DSA distances between them. This produced a "null distribution" of DSA scores that come from comparing identical dynamical systems, for which we report the 95% CI below (Header labels, e.g. "Level 1" for task complexity match the x‑axis levels used in the main‑text figures). The CIs below are on average an order of magnitude smaller than the DSA distance across networks. Therefore, it is valid to consider networks trained from different initialization as having statistically significantly different neural dynamics.
>
> | Task Complexity| Level 1 | Level 2 | Level 3 | Level 4 |
> |---|---|---|---|---|
> | N-BFF | [0.011, 0.013] | [0.009, 0.016] | [0.008, 0.013] | [0.006, 0.009] |
> | Delayed Discrim. | [0.039, 0.064] | [0.014, 0.076] | [0.025, 0.032] | | |
> | Sine Wave Gen | [0.057, 0.102] | [0.054, 0.081] | [0.048, 0.073] | | |
> | Path Integr. | [0.023, 0.037] | [0.010, 0.018] | | | | |
>
> * For PIF distance, we similarly establish a noise floor by randomly permuting the trained networks' recurrent matrix and computing the distance between the permuted weight matrix and the original one. The PIF metric reliably recovers a PIF distance of 0.
>
> &nbsp;
>
> > Q2: Absence of flow field visualizations to confirm qualitative differences in neural dynamics.
>
> Thank you for the suggestion! Unfortunately, the change in the rebuttal rules does not allow us to upload figures, but we do find cases where the dynamics look qualitatively different and will include these visualizations in the revised supplemental materials. That said, we believe visualization is not always the most reliable way to detect differences in neural dynamics, because:.
>
> * **Dynamical differences may lie in high-dimensional space and be
> visually subtle.** As suggested by [56], networks trained on the same task can develop the same dynamical objects and transitions between them. We therefore aim to go beyond looking at the presence of dynamical objects (fixed points, limit cycles, etc.) and their transitions, and instead capture neural dynamics via a high‑dimensional linear operator with DSA, directly quantifying differences in the learned dynamics operators. Moreover, we found it hard to compare dynamics in visualizations beyond 3D, and the dimensionality of the neural activity of our tasks often exceeded that, as reported below for Q3.
>
> * **Visible differences can reflect geometry, not topology.** Flow fields and phase portraits primarily reflect the geometry of neural activity, while we define “similar” neural dynamics as topologically conjugate neural dynamics. Moreover, even if two networks have topologically equivalent dynamics, they can be related by a rotation, making their flow fields look very different (see, for example, Figure 6 in the appendix of [67]).
>
> * **We do observe qualitative differences in some cases.** In two‑channel Delayed Discrimination, networks develop two sets of fixed points (one per channel) representing the previous stimulus during the delay period. The PCA plots reveal either factorized, disentangled versus entangled fixed points for different channels in the state space. Linking this entanglement to behavioral/OOD generalization would be a promising next step to connect dynamical and behavioral degeneracy.
>
> &nbsp;
>
> > Q3: How do the measures and trends extend to chaotic or high-dimensional RNNs? For instance, are the same trends visible with chaotic attractors?
>
> We remark that our original task suite comprises neuroscience‑motivated tasks that can produce high‑dimensional neural dynamics as the number of input–output channels increase. We see that the average number of principal components explaining 95% variance are: N‑BFF (3/8/16/32 channels: 5, 10.32, 14.82, 29.2), Delayed Discrimination (2/3/4 channels: 21.5, 42.08, 42.1), Sine‑Wave Generation (2/3/4 channels: 17.38, 22.9, 29), Path Integration (2/3 channels: 5.86, 7.26).
>
> We now **added a chaotic attractor task and verified that the effects of all four factors on dynamical and weight degeneracy are consistent with Table 1**. We will add full details  to the revised supplementary materials. Meanwhile, we summarize our results below:
>
> We implemented the Lorenz 96 model [1] with 16, 24, and 32 dimensions and trained 50 RNNs on next-step prediction until the networks achieve a near asymptotic MSE loss at 0.0005. After training, the average Lyapunov exponent of RNNs trained on the Lorenz 96 attractor with 16 dimensions is $12.58 \pm 0.74$, indicating chaotic neural dynamics.
>
> | Task Complexity |N=16 | N=24 | N=32 |
> |---|---|---|---|
> | Dynamical Degeneracy | 2.58e-01 ± 7.02e-05 | 2.57e-01 ± 6.97e-05 | 2.52e-01 ± 6.03e-05 |
> | Weight Degeneracy | 9.68e-04 ± 9.39e-07 | 1.02e-03 ± 1.14e-06 | 1.14e-03 ± 1.76e-06 |
>
> | Feature Learning Strength |$\gamma=4$ | $\gamma=5$ | $\gamma=10$ | $\gamma=20$ |
> |---|---|---|---|---|
> | Dynamical Degeneracy | 2.79e-01 ± 1.68e-04 | 2.77e-01 ± 1.74e-04 | 2.75e-01 ± 1.49e-04 | 2.70e-01 ± 2.16e-04 |
> | Weight Degeneracy | 1.35e-01 ± 1.08e-04 | 1.42e-01 ± 1.04e-04 | 1.76e-01 ± 1.24e-04 | 2.25e-01 ± 2.13e-04 |
>
> | Network Width |128 units | 256 units | 512 units |
> |---|---|---|---|
> | Dynamical Degeneracy | 2.80e-01 ± 1.39e-04 | 2.66e-01 ± 1.09e-04 | 2.55e-01 ± 1.04e-04 |
> | Weight Degeneracy | 1.23e-01 ± 8.82e-05 | 8.79e-02 ± 3.30e-05 | 6.23e-02 ± 1.21e-05 |
>
> | Rank Regularization |$\lambda=0$ | $\lambda=5e-5$ | $\lambda=1e-4$ | $\lambda=5e-4$ |
> |---|---|---|---|---|
> | Dynamical Degeneracy | 2.58e-01 ± 7.02e-05 | 2.10e-01 ± 1.12e-04 | 2.02e-01 ± 9.60e-05 | 1.98e-01 ± 1.35e-04 |
> | Weight Degeneracy |  9.68e-04 ± 9.39e-07 | 7.27e-05 ± 4.71e-07 | 1.51e-05 ± 3.71e-07 | 1.74e-06 ± 6.91e-09 |
>
> Our original tasks already span fixed‑point, limit‑cycle, and attractor manifold regime; the new chaotic regime extends this coverage.
>
>
> &nbsp;
>
> > Q4: In Figure 3, are network sizes held constant across varying complexities?
>
> Yes, the network sizes are held constant (128 units) across varying complexities, and indeed increasing task complexity forces equally-sized networks to utilize higher capacity . It is natural to assume that this would reduce the degeneracy in their solutions. This is argued by Cao & Yamins (2021) proposing the Contravariance Principle, but not rigorously tested with experiments. Our findings suggest that when networks are forced to operate at higher capacity, only behavioural and dynamical degeneracy decreases, while weight degeneracy increases. Hence, we reveal a more complex, unintuitive relationship between task complexity and degeneracy of solution. And we believe that the computational neuroscience community (and anyone trying to reverse engineer RNNs) should be aware of this.
>
> > Q5: How well do these RNNs actually solve the tasks?
>
> We applied an early stopping to all networks trained on the same task, regardless of the task complexity. The early stopping threshold is a mean-squared error (MSE) of 0.001 for N-BFF, 0.01 for delayed discrimination, and 0.05 for Sine Wave Generation and Path Integration. The dynamic range for Path Integration is [-5, 5] and [-1, 1] for the other tasks. We confirmed that all networks used for analysis in the paper reached this threshold, and groups of networks trained on the same task all achieve **comparable loss** regardless of their levels of degeneracy. We will explicitly report the model performance in the revised paper, and add figures to the supplementary material showing example trials that juxtapose the ground truth and predicted signals.
>
> > Q6: The paper refers to dispersion of local minima but does not show empirical support.
>
> At line 177, we present this as speculation (not a main claim). We apologize for the confusion and will move it to the Discussion section. It is also possible that the increased weight degeneracy is caused by wider basins in a large solution space. Clarifying the cause of increased weight degeneracy for harder tasks, or providing a mechanistic link between loss‑landscape geometry and weight degeneracy, remains an exciting direction for future work, but is beyond the scope of the current paper.
>
> > Q7: Are DSA comparisons based on autonomous (free-running) dynamics or input-driven trajectories?
>
> Thank you for raising this important point! In the original DSA paper, the authors demonstrate the use of DSA using either autonomous systems or systems with controlled, low-dimensional inputs. We believe that we are performing our experiments in exactly the same regime as the original paper, since our tasks involve either autonomous dynamics or use simple controlled inputs. We also take care to ensure that the inputs being used to compare networks are exactly matched.
>
> &nbsp;
> ### References
> [1] E. N. Lorenz, “Predictability: a problem partly solved,” in *ECMWF Seminar on Predictability*, 4–8 Sept. 1995. Reading, U.K.: European Centre for Medium-Range Weather Forecasts, 1996.

---

> > ### Comment · Reviewer_fDhN · 2025-08-02
> >
> > I thank the authors for their thorough rebuttal and the new results which addressed most of my concerns.
> >
> > I trust that the authors will include the promised extensions in the revised paper.
> >
> > Everything has been clarified except one thing. I still feel the number of datapoints is rather small (2–4 data points per task variant in e.g. Figs. 3,6,7) which in my mind weakens broad generalization claims. Could you please comment on that.

---

> ### Author Response · Authors · 2025-08-04
>
> Thank you for the follow-up! We agree it is important to know whether the degeneracy trends generalize to intermediate values and beyond the ranges reported in the paper. To test this, we used N-BFF as an example and ran a **dense sweep** both interpolating within and extrapolating beyond the ranges shown in Figs. 3 and 6–8. Specifically:
>
> - Feature-learning strength: 20 intermediate $\gamma$ values, uniformly spaced in $[0.1, 4.0]$.
> - Network size: 12 intermediate widths from 32 to 512 recurrent units, in increments of 16 or 32.
> - Rank regularization: 15 intermediate $\lambda$ values; five from $5{\times}10^{-7}$ to $9{\times}10^{-7}$ (step $10^{-7}$), and ten from $10^{-6}$ to $5{\times}10^{-5}$ (step $10^{-6}$).
>
> Across **dynamical, weight, and behavioral** levels, the **degeneracy trends remain unchanged** and **interpolate smoothly** across these intermediate values. Due to space limits, we include 10 intermediate values per factor in the tables below and will place full details in the supplemental materials in the revised paper.
>
> For task complexity, we ran a dense sweep from 3BFF to 64BFF and are still waiting for the experiments to finish. We expect to post the results before the Author–Reviewer Discussion period ends, so stay tuned!
>
> As context, prior work commonly studies the single-channel Delayed Discrimination, single-output-channel Sine-Wave Generation, and 2D Path Integration due to their **trainability** and **biological relevance** [1-4]. Pushing to substantially higher complexities (e.g., Path Integration in 4D space) to include more datapoints markedly increases optimization difficulty and could reduce biological plausibility, thus limiting their relevance for task-driven modeling community in neuroscience; for those reasons, in Fig. 3,  we used 2D and 3D Path Integration as the complexity sweep for that task.
>
> We also remark that each plotted point in Figs. 3 and 6–8 aggregates **50 independently trained networks** and **all pairwise comparisons** between them, yielding ~**85,000** distance computations at each of the dynamical and weight levels. Importantly, for each factor, the **same qualitative patterns** on degeneracy replicate across **four tasks**, so our claims do not hinge on any single panel.
>
> &nbsp;
>
> | Feature Learning Strength ($\gamma$) |0.1 | 0.2 | 0.4 | 1.0 | 1.6 | 2.0 | 2.4 | 2.8 | 3.2 | 3.6 |
> |---|---|---|---|---|---|---|---|---|---|---|
> | Dynamical Similarity | 7.707e-02 ± 2.049e-04 | 7.647e-02 ± 1.984e-04 | 7.084e-02 ± 1.936e-04 | 6.645e-02 ± 1.727e-04 | 6.536e-02 ± 1.775e-04 | 6.221e-02 ± 2.464e-04 | 6.013e-02 ± 2.625e-04 | 5.780e-02 ± 2.655e-04 | 5.617e-02 ± 2.858e-04 | 5.586e-02 ± 3.318e-04 |
> | Weight Degeneracy | 1.246e-01 ± 9.299e-05 | 1.250e-01 ± 9.762e-05 | 1.255e-01 ± 9.668e-05 | 1.275e-01 ± 9.576e-05 | 1.280e-01 ± 9.392e-05 | 1.290e-01 ± 9.623e-05 | 1.301e-01 ± 9.749e-05 | 1.313e-01 ± 1.015e-04 | 1.325e-01 ± 1.021e-04 | 1.338e-01 ± 1.053e-04 |
> | Behavioral Degeneracy | 1.474e-02 | 1.434e-02 | 1.208e-02 | 1.082e-02 | 1.042e-02 | 9.233e-03 | 8.874e-03 | 8.794e-03 | 8.398e-03 | 5.943e-03 |
>
> &nbsp;
>
> |Network width |32 | 48 | 64 | 128 | 160 | 192 | 224 | 256 | 300 | 512 |
> |---|---|---|---|---|---|---|---|---|---|---|
> | Dynamical Similarity | 7.230e-02 ± 1.663e-04 | 7.009e-02 ± 1.580e-04 | 6.931e-02 ± 1.717e-04 | 6.686e-02 ± 2.063e-04 | 6.637e-02 ± 2.089e-04 | 6.462e-02 ± 2.008e-04 | 6.432e-02 ± 2.197e-04 | 6.363e-02 ± 2.141e-04 | 6.302e-02 ± 2.115e-04 | 6.153e-02 ± 2.858e-04 |
> | Weight Degeneracy | 2.594e-01 ± 6.898e-04 | 2.092e-01 ± 4.027e-04 | 1.808e-01 ± 2.595e-04 | 1.266e-01 ± 9.506e-05 | 1.127e-01 ± 6.349e-05 | 1.028e-01 ± 4.645e-05 | 9.510e-02 ± 4.215e-05 | 8.891e-02 ± 2.966e-05 | 8.203e-02 ± 2.978e-05 | 6.265e-02 ± 1.216e-05 |
> | Behavioral Degeneracy | 1.198e-02 | 1.271e-02 | 1.129e-02 | 9.521e-03 | 7.520e-03 | 6.876e-03 | 5.412e-03 | 6.460e-03 | 6.218e-03 | 4.954e-03 |
>
> &nbsp;
>
> | Rank Regularization ($\lambda$) |5e-7 | 7e-7 | 9e-7 | 2e-6 | 4e-6 | 5e-6 | 6e-6 | 7e-6 | 8e-6 | 1e-5 |
> |---|---|---|---|---|---|---|---|---|---|---|
> | Dynamical Similarity | 7.179e-02 ± 2.438e-04 | 7.174e-02 ± 2.486e-04 | 7.171e-02 ± 2.445e-04 | 7.155e-02 ± 2.407e-04 | 7.074e-02 ± 2.519e-04 | 7.102e-02 ± 2.544e-04 | 7.007e-02 ± 2.571e-04 | 6.966e-02 ± 2.619e-04 | 6.862e-02 ± 2.443e-04 | 6.564e-02 ± 2.302e-04 |
> | Weight Degeneracy | 9.194e-04 ± 1.047e-06 | 9.190e-04 ± 9.708e-07 | 9.181e-04 ± 9.787e-07 | 9.124e-04 ± 1.032e-06 | 8.987e-04 ± 9.966e-07 | 8.894e-04 ± 1.078e-06 | 8.760e-04 ± 1.051e-06 | 8.599e-04 ± 1.007e-06 | 8.385e-04 ± 1.146e-06 | 7.701e-04 ± 1.356e-06 |
> | Behavioral Degeneracy | 2.277e-04 | 1.868e-04 | 1.819e-04 | 1.800e-04 | 1.457e-04 | 1.252e-04 | 9.834e-05 | 1.005e-04 | 6.359e-05 | 5.290e-05 |
>
> &nbsp;
> ### References
> [1] N. Maheswaranathan et al., NeurIPS 2019
> [2] E. Turner, K. Dabholkar, and O. Barak, NeurIPS 2021
> [3] C. J. Cueva and X.-X. Wei, ICLR 2018
> [4] A. Banino et al., Nature 2018

---

> > ### Comment · Reviewer_fDhN · 2025-08-04
> >
> > I appreciate the authors’ comprehensive reply and the additional results. As all of my concerns have now been addressed, I am happy to increase my review score.

---

### Official Review · Reviewer_vngU · 2025-07-03

**Clarity:** 3
**Significance:** 3
**Originality:** 3
**Rating:** 4
**Confidence:** 4

**Summary:**

This paper proposes a unified framework to measure degeneracy in RNNs across three dimensions: behavioral, weight, and dynamic degeneracy. It quantifies the degree of degeneracy as a function of various factors, including task complexity, network size, learned features, and regularization.

**Questions:**

- Can the authors demonstrate that all models of interest converge to similar losses (e.g., within 2% of the average)? How robust are the results if this threshold is varied (e.g., within 5% of the average)?

- The paper defines network size in terms of width, but depth also contributes to model capacity. Do the relationships reported in Table 1 hold for networks of varying depth?

We may consider increasing the score if these questions are addressed.

**Ethical Concerns:**

["NO or VERY MINOR ethics concerns only"]

**Final Justification:**

This work investigates the degeneracy of RNNs across different dimensions and common factors shared across tasks. The rebuttal and discussion have effectively addressed my concerns regarding training convergence, the impact of network depth, and the variability across trained models. I recommend accepting this work and raise my score to 4.

**Limitations:**

Yes

**Quality:**

2

**Strengths And Weaknesses:**

Strengths:
- The authors propose three metrics to quantify degeneracy across behavioral, weight, and dynamic dimensions.
- The paper analyzes multiple factors and establishes qualitative relationships between these factors and different types of degeneracy.

Weaknesses:
- The theoretical (even qualitative) rationale behind the observed relationships is limited.
- The degeneracy measurements assume that different networks achieve comparable training losses, yet no explicit justification for this assumption is provided.

---

> ### Author Rebuttal · Authors · 2025-07-30
>
> We thank the reviewer for the thoughtful questions!
>
> > [Q1, W2]: Can the authors demonstrate that all models of interest converge to similar losses? How robust are the results if this threshold is varied?
>
> * In all experiments, we train networks until them reach a **near‑asymptotic, task‑specific mean-squred error (MSE) threshold** (0.001 for N-BFF, 0.01 for Delayed Discrimination, and 0.05 for Sine‑Wave Generation and Path Integration), after which we allow a patience period of 3 epochs and stop training to measure degeneracy. This early‑stopping criterion ensures that networks trained on the same task/condition achieve comparable final losses before any degeneracy analysis. While we applied this protocol throughout, we recognize it was not stated explicitly in the paper; in the revision we will describe it explicitly in Section 2.1 Model architecture and training procedure section.
>
> * To quantify the residual variation, we report the coefficient of variation (CV) of the final training loss across seeds for each condition, expressed as % of the mean. Header labels match the x‑axis levels used in the main‑text figures. Final losses cluster tightly near small values of the loss threshold, so even a double-digit CV translates to very small absolute variation. For example, a 10% CV at an MSE of 0.001 implies an s.d. of $1 \times 10^{-4}$; at 0.01 it’s $1 \times 10^{-3}$.
> We further note that these variability values are not monotonic in any factor and sometimes move opposite to the degeneracy trends, arguing against a loss‑dispersion confound.
>
> | Task Complexity |Level 1| Level 2 | Level 3| Level 4|
> |---|---:|---:|---:|---:|
> |N‑BFF|6.30%|4.60%|9.30%|3.50%|
> |Delayed Discrim.|15.90%|8.40%|9.50%||
> |Sine Wave Gen.|9.94%|9.20%|8.70%||
> |Path Integr.|9.16%|2.85%|||
>
> | Feature Learning Strength |$\gamma_1$|$\gamma_2$|$\gamma_3$|$\gamma_4$|
> |---|---:|---:|---:|---:|
> |N‑BFF|9.70%|9.10%|13.40%|11.70%|
> |Delayed Discrim.|8.70%|12.60%|11.70%|12.30%|
> |Sine Wave Gen.|3.50%|3.90%|10.90%|11.70%|
> |Path Integr.|5.40%|5.20%|6.20%||
>
> | Network Width| 64 units|128 units|256 units|
> |---|---:|---:|---:|
> |N‑BFF|3.80%|4.20%|3.50%|
> |Delayed Discrim.|3.30%|3.00%|3.20%|
> |Sine Wave Gen.|17.80%|16.60%|16.40%|
> |Path Integr.|5.10%|5.40%|5.90%|
>
> | L1 Regularization |$\lambda_1$|$\lambda_2$|$\lambda_3$|$\lambda_4$|
> |---|---:|---:|---:|---:|
> |N‑BFF|2.10%|6.90%|1.10%||
> |Delayed Discrim.|15.90%|14.50%|16.70%|14.90%|
> |Sine Wave Gen.|10.40%|11.10%|11.10%||
> |Path Integr.|9.00%|7.10%|3.00%||
>
>
> | Rank Regularization |$\lambda_1$|$\lambda_2$|$\lambda_3$|$\lambda_4$|
> |---|---:|---:|---:|---:|
> |N‑BFF|2.10%|7.20%|4.30%||
> |Delayed Discrim.|15.90%|16.90%|13.60%|12.10%|
> |Sine Wave Gen.|13.90%|14.30%|15.90%||
> |Path Integr.|7.70%|7.90%|6.70%||
>
> * **Degeneracy trends for a looser stopping criterion on N-BFF, consistent with Table 1.** We verified that using a slightly looser stopping criterion for N‑BFF (MSE = 0.005 instead of 0.001) leaves the degeneracy trend for all four factors unchanged, consistent with Table 1. We also computed the CV of final training losses and confirmed that the higher threshold yields slightly greater across‑seed variability, yet the degeneracy trends are robust to both the magnitude and variability of the final loss. Due to the character limit, we show degeneracy trend at this higher loss threshold for varying feature learning strength, network size, and rank regularization an examples. We will include the full analysis for all four tasks in the supplementary material of the revised manuscript.
>
> | Feature Learning Strength| $\gamma=0.5$ | $\gamma=1$ | $\gamma=2$ | $\gamma=3$ |
> |---|---|---|---|---|
> | Dynamical Degeneracy | 5.51e-02 ± 6.08e-05 | 5.27e-02 ± 6.98e-05 | 4.73e-02 ± 7.92e-05 | 4.54e-02 ± 8.83e-05 |
> | Weight Degeneracy | 1.25e-01 ± 1.00e-04 | 1.26e-01 ± 9.52e-05 | 1.28e-01 ± 9.78e-05 | 1.31e-01 ± 1.01e-04 |
>
> | Network Width | Width=64 | Width=128 | Width=256 |
> |---|---:|---:|---:|
> | Dynamical Degeneracy | 4.29e-02 ± 8.41e-05 | 4.11e-02 ± 8.04e-05 | 3.94e-02 ± 8.48e-05 |
> | Weight Degeneracy    | 1.81e-01 ± 2.74e-04 | 1.27e-01 ± 1.08e-04 | 8.90e-02 ± 3.79e-05 |
>
> | Rank Regularization |$\lambda=0$ | $\lambda=1e-06$ | $\lambda=5e-06$ |
> |---|---|---|---|
> | Dynamical Degeneracy | 1.22e-01 ± 1.58e-04 | 1.21e-01 ± 1.54e-04 | 1.15e-01 ± 1.78e-04 |  |
> | Weight Degeneracy | 9.49e-04 ± 8.66e-07 | 9.42e-04 ± 8.50e-07 | 8.78e-04 ± 8.08e-07 |  |
>
>
> * We also remark that precisely matching final losses and their variability across runs is inherently noisy; moreover, post‑hoc filtering to include only runs within a tight loss band can introduce selection bias (e.g., favoring certain optimization paths or initializations). Our method motivates future theoretical analysis where the variability of the loss can be controlled exactly, to provide a theoretical relationship between loss dispersion and solution degeneracy. We thank you for your insightful question again!
>
>
> &nbsp;
>
> > Q2: The paper defines network size in terms of width, but depth also contributes to model capacity. Do the relationships reported in Table 1 hold for networks of varying depth?
>
>
> We thank the reviewer for raising the important point of examining the network depth effect on solution degeneracy. Below we position the effect of depth on degeneracy within the scope of our contributions, clarify our design choice, and report the requested experiment.
>
> * Our study is motivated by task‑driven modeling in neuroscience, where researchers reverse‑engineer RNNs trained on neuroscience/cognitive tasks to understand neural computation. In this literature, recurrent models are predominantly **single‑layer (depth‑1), vanilla RNNs** [5, 18, 57, 72, 76, 79]; increasing depths would imply a very specific **biological** architectural bias that has not been explored much, to the best of our knowledge. More biologically grounded alternatives are multi‑region RNNs, which can be viewed as a single‑layer RNN with structured/sparse connectivity. While we focus on RNNs due to their relevance for modeling neural computations in our paper, depth is of fundamental interest to **deep feedforward networks** (e.g. CNNs). In feedforward networks, computations are carried out in terms of a limited number of transformations of a model’s representations over it's layers, instead of dynamics. Therefore, alternative metrics should be used to measure degeneracy (representational instead of dynamical degeneracy). Lots of work has been conducted to study the representations across depth in feedforward networks [1-3]. To the best of our knowledge the specific relation between solution degeneracy and model depth has not been studied yet, and could be a promising future direction!
>
> * **Controlling feature learning to isolate the effect of depth in deep RNNs requires a substantial new theory**. To isolates the network width effect on solution degeneracy, we use μP to equalize feature‑learning strength across width, but a depth‑consistent parameterization that achieves the same control for deep RNNs is not available and, to our knowledge, remains an open problem. To the best of our knowledge, only one recent paper [4] attempts a scaling limit for infinite depth, and it does so for feedforward residual networks with additional constraints (including effectively width=1), explicitly noting the difficulty of obtaining a depth‑consistent scaling for general width and nonlinear settings. Given these constraints, we made the conscious choice to keep depth fixed and vary width. We will update the paper to explicitly refer to width when we discuss model size.
>
> * We ran the requested experiment, and found **that dynamical and weight degeneracy are in general lower for deeper RNNs, although not perfectly monotonic.** We varied the number of recurrent layers from 1 to 4, while keeping the width for each layer fixed at 128 units, and examined dynamical and weight degeneracy on the Sine Wave Generation task. Overall, degeneracy decreases for deeper networks, consistent with Table 1. However, the trend is not perfectly monotonic. Upon closer examination of the feature learning strength of networks with different depth, we found that the feature learning strength is indeed unstable across depth, which could confound the change in degeneracy.
>
> | Model Depth |1 | 2 | 3 | 4 |
> |---|---|---|---|---|
> | Dynamical Degeneracy | 2.33e-01 ± 1.22e-04 | 2.35e-01 ± 2.20e-04 | 2.32e-01 ± 5.72e-05 | 2.30e-01 ± 5.66e-05 |
> | Weight Degeneracy | 1.10e-03 ± 2.17e-06 | 1.06e-03 ± 2.12e-06 | 1.06e-03 ± 1.76e-06 | 1.06e-03 ± 3.00e-06 |
>
> &nbsp;
>
> > W1: The theoretical rationale behind the observed relationships is limited.
>
> We agree that a theoretical account of the relationships between the four factors and degeneracy would be valuable. Our contribution is a systematic empirical framework for quantifying solution degeneracy across behavior, dynamics, and weight space. We establish robust empirical relationships between solution degeneracy and task complexity, learning regime, network width, and regularization. The consistency of results across tasks strongly motivates future theoretical work.
>
>
> We hope our responses address your questions and concerns. If any further details or clarifications would be helpful, please let us know!
>
> &nbsp;
> ### References
> [1] M. Raghu, J. Gilmer, J. Yosinski, and J. Sohl-Dickstein, “SVCCA: Singular Vector Canonical Correlation Analysis for deep learning dynamics,” NeurIPS, 2017.
>
> [2] F. Chen, D. Kunin, A. Yamamura, and S. Ganguli, “Stochastic Collapse: How Gradient Noise Attracts SGD Dynamics Towards Simpler Subnetworks,” NeurIPS, 2023.
>
> [3] F. Cagnetta, L. Petrini, U. M. Tomasini, A. Favero, and M. Wyart, “How Deep Neural Networks Learn Compositional Data: The Random Hierarchy Model,” Phys. Rev. X, vol. 14, no. 3, p. 031001, 2024. doi: 10.1103/PhysRevX.14.031001.
>
> [4] G. Yang, D. Yu, C. Zhu, and S. Hayou, “Tensor Programs VI: Feature Learning in Infinite-Depth Neural Networks,” ICLR, 2024.

---

> > ### Comment · Reviewer_vngU · 2025-08-08
> >
> > Thank the authors for their thorough rebuttal. Most of my questions have been addressed, and I trust they will include the additional results in the revised manuscript. However, I still have one concern: the coefficients of variation (CVs) for some tasks exceed 16%. With that degree of variability, I am not convinced the RNNs can be considered similar, as a 16% CV in loss may ultimately influence performance. Please comment on this point.

---

> ### Author Response · Authors · 2025-08-05
>
> Dear Reviewer vngU,
>
> Since the discussion period is closing in a few days, we wanted to check in to see if our responses properly addressed your concerns. We’d appreciate hearing your feedback to ensure we've fully clarified the point. Thank you again for your thoughtful reviews!

---

> ### Author Response · Authors · 2025-08-09
>
> Thanks for the follow-up! We address the concern as follows:
>
> - We would like to remark that **we focus on the variability of the out-of-distribution (OOD) performance** (standard deviation of the mean-squared error), **instead of the performance itself as a measure of behavioral degeneracy**.
>
> - The mean and the variability of the **training performance and OOD performance are decoupled**, suggesting that our main claims on **behavioral degeneracy are not confounded by the specific level of training variability**. Here, we show an example of networks trained with different feature learning strengths (i.e. different $\gamma$) on the Delayed Discrimination task. The mean of the training loss is highly non-monotonic with respect to $\gamma$, while OOD loss is positively correlated with $\gamma$ . The standard deviation (SD) of the training loss is also non-monotonic in $\gamma$, while the SD of the OOD loss (our measure of behavioral degeneracy) consistently increases for larger $\gamma$. Therefore, **the trend in behavioral degeneracy is not a function of the training loss variability, ruling out the possibility that the training loss variability influences our claims on behavioral degeneracy**. On the dynamical and weight level, we see that those measures are not confounded by the training loss variability either. In fact, the CV of final losses reported in the previous comment are not monotonic in any factor and sometimes move opposite to the degeneracy trends.
>
>   **Training loss**
>   | $\gamma$ |1 | 2 | 3 | 4 |
>   |---|---|---|---|---|
>   | mean | 9.67e-04 | 9.14e-04 | 9.37e-04 | 9.10e-04 |
>   |  std  | 8.43e-05 | 1.15e-04 | 1.10e-04 | 1.12e-04 |
>   |  cv   | 0.087 | 0.126 | 0.117 | 0.123 |
>
>   **OOD loss**
>   | $\gamma$ |1 | 2 | 3 | 4 |
>   |---|---|---|---|---|
>   | mean | 1.26e-02 | 1.19e-02 | 1.30e-02 | 1.58e-02 |
>   |  std | 2.55e-03 | 2.70e-03 | 3.25e-03 | 3.62e-03 |
>   |  cv  | 0.202 | 0.227 | 0.249 | 0.229 |
>
>
> - **Degeneracy trends are robust to the level of training loss variability**. Across all four tasks, despite some tasks having less variable training loss variability (<5%), while others have slightly higher variability (>10%), the trends in behavioral, dynamical, and weight degeneracy hold very consistently across tasks, demonstrating its robustness across loss variability. Moreover, as shown in our earlier rebuttal experiments on using a higher loss threshold for N-BFF task, within the same task, the degeneracy trends are also robust to reasonable differences in training loss variability.
>
> - **Even though the CV of final losses sometimes looks large, the networks converged well on a global scale**. Across our results, the mean MSE after training is under 2% of the mean MSE at initialization, indicating that training has converged well. Individual values: 0.059% (N-BFF), 1.6% (Delayed Discrimination), 0.32% (Sine-Wave Generation), 0.94% (Path Integration). CV can look large when the mean is tiny (the denominator is small). For example, a 16% CV on Sine-Wave Generation corresponds to ~0.05% of the initialization loss, which is consistent with minor differences because of stochastic gradients rather than meaningful under-training.
>
> - **Our convergence protocol is stricter than common practice in task-driven modeling**. Prior work comparing “similar” RNNs often uses a fixed number of epochs and treats networks as “similar” once losses fall below a threshold (e.g., Maheswaranathan et al., NeurIPS 2019: fixed 2000 epochs on Sine-Wave Generation; Turner et al., NeurIPS 2021: fixed 1000 epochs on Delayed Discrimination; Ostrow et al., NeurIPS 2023: networks with loss < 0.05 on 3BFF are considered equivalent). To illustrate how fixed-length training used in previous work can yield high variability in final losses, we trained Sine-Wave Generation with 2/3/4 input-output channels for 2000 epochs (no early stopping). The mean final training losses is around ~0.01 but with a huge CV across seeds. In our work, we apply early stopping to control loss variability across seeds; in some cases our stopping threshold is **50× tighter than prior work** (e.g., 0.001 on N-BFF vs. 0.05 in Ostrow et al., 2023). We use a 3-epoch patience window after hitting the threshold; while this can introduce slight residual variability, it helps ensure models have truly converged.
>
>   | Sine Wave Gen. |2 channels | 3 channels | 4 channels|
>   |---|---|---|---|
>   | CV | 0.793 | 0.385 | 0.553 |
>
> - **There is no universal CV cutoff as the train-loss variability also reflects task-specific loss-landscape geometry.**
>
> We hope the above addresses the concern. If so, we invite the reviewer to consider increasing their evaluation score. Thank you for our thoughtful review and questions again!

---

> > ### Comment · Reviewer_vngU · 2025-08-09
> >
> > Thanks for the clarification regarding the CV measurements and training convergence. I have no further concerns. I suggest including the CV measurements in the appendix. The paper presents interesting results and solid experiments on measuring RNN degeneracy. I will raise my score.

---

### Official Review · Reviewer_VYSY · 2025-07-03

**Clarity:** 4
**Significance:** 3
**Originality:** 3
**Rating:** 5
**Confidence:** 5

**Summary:**

This paper studies solution degeneracy in vanilla recurrent neural networks (RNNs) across three levels: behavior, neural dynamics, and weight space. The paper runs a systematic empirical analysis by training ~50 networks on each of four neuroscience inspired tasks, varying: task complexity, feature learning, network size, and regularization. Previous work has mostly studied trained networks after fixing these variables, so this systematic study is quite timely.

The paper finds interesting relationships between the behavioral, dynamical, and weight space similarities of trained networks as a function of the task complexity, network size, and regularization. In particular, it finds that for some features (such as task complexity), the dynamical degeneracy and the weight degeneracy move in opposite directions ("contravariant") while for other features (such as regularization), they move in the same direction ("covariant").

Overall, the paper provides a clear summary of the effect of key architectural choices on RNN solution degeneracy, which both can serve as a practical guide for neuroscientists using RNNs as well as opening the door for future studies to better explore the solution space of RNNs on nonlinear tasks.

**Questions:**

1. Do you think the results would hold for gated RNNs?
2. Do you normalize the weight-change norm for the number of steps (T) required for convergence? If not, does incorporating some normalization there affect the results (I'm not sure what the right normalization is, maybe divide by $\sqrt(T)$?)
3. What about $\ell_2$ regularization? I'm assuming it has a similar effect as $\ell_1$ and low-rank penalties. I think it would be helpful to add $\ell_2$ regularization experiments in section 3.5 for completeness.
4. In figure 2B, for the sine-wave generation task, are the colored dots fixed points or samples from trajectories? I would assume the latter. If so, it might be helpful to show the fixed points as points and the trajectories as lines, to be consistent with the other panels.

**Ethical Concerns:**

["NO or VERY MINOR ethics concerns only"]

**Final Justification:**

After reading the other reviews and the author rebuttal, I am keeping my score at 5, but I still think this is a solid paper and am in favor of acceptance.

**Limitations:**

yes

**Quality:**

4

**Strengths And Weaknesses:**

# Strengths
- The paper studies a critical and (as pointed out in the introduction) relatively understudied problem of solution degeneracy in RNNs.
- The paper is really well written, with clear motivation and thorough descriptions of the experiments and results.
- The inclusion of both code and the trained model weights will be a valuable resource for the community.
- The study examines networks trained across four disparate (but all strongly motivated in neuroscience) tasks. The consistency of the results across these tasks really strengthens the likely generalizability of the findings.

# Weaknesses
- Narrow definition of behavioral degeneracy: The behavioral degeneracy is defined as out-of-distribution generalization to longer time sequences (temporal generalization). However, this represents only a narrow slice of possible generalization we might expect networks to exhibit. Calling this category "behavioral degeneracy" slightly overstates the claims, in my opinion. I think being explicit about this and referring to this category as "temporal generalization" instead of "behavioral degeneracy" would be a more accurate description. (Or, run additional experiments to examine other types of generalization: such as generalization to noise, perturbations, or other types of task variability).
- Limitation to vanilla RNNs. The study exclusively studies vanilla RNNs with tanh nonlinearities, which lack the gating mechanisms of LSTMs or GRUs. It is unclear whether the findings would generalize to gated networks.
- Potential confound in the weight change norm: When quantifying the weight-change norm (Section 3.2.1), do you control for the number of training steps required for each task? I would imagine that due to the stochastic nature of SGD, the change in weight norm trivially grows with the number of steps (e.g. as discussed in https://arxiv.org/pdf/1806.08805). So if the number of steps varies across tasks, the weight change norm will also, in a way that has nothing to do with weight degeneracy.

---

> ### Author Rebuttal · Authors · 2025-07-30
>
> Thank you for the thoughtful, constructive review and suggestions! We are glad you find the study of solution degeneracy critical and timely. Below we address the comments you raised in Questions and Weaknesses part of the review.
>
> ### Questions
>
> > [Q1, W2]: Do you think the results would hold for gated RNNs?
>
> We agree that testing if results generalize to gated RNNs is an important question! We do think the trend in dynamical degeneracy will hold with gated RNNs, as pointed out by [1], architectural choices affect the geometry but not topology of neural dynamics, which is shaped by the task structure. At the same time, the DSA metric we used to quantify dynamical degeneracy compares the topological structure of neural dynamics while being invariant to changes in geometry. As a preliminary investigation into the results’ generalizability with respect to architectural choices, we have trained GRUs to perform the Sine Wave Generation task, while changing task complexity by varying number of input-output channels. As task complexity increases, the dynamical degeneracy drops while weight degeneracy rises, consistent with what we observed for vanilla RNNs. We will add this result to the supplementary material in the revised version of the paper.
>
> | Task Complexity |1 channels | 2 channels | 3 channels |
> |---|---|---|---|
> | Dynamic | 2.61e-01 ± 2.09e-04 | 2.07e-01 ± 2.09e-04 | 1.86e-01 ± 7.89e-05 |
> | Weight | 1.25e-03 ± 1.16e-05 | 2.82e-03 ± 3.17e-05 | 4.43e-03 ± 2.71e-05 |
>
> > [Q2, W3]: Do you normalize the weight-change norm for the number of steps (T) required for convergence?
>
> * We plotted the weight change norm of our networks after every gradient step, and found it plateaued early in training, and the $||\Delta W||$ between consecutive gradient steps fell below $10^{-7}$ per weight towards end of training, which can be considered neglectable compared to the weight change norm on the order of $10^{-3}$ per weight. We therefore do not normalize the weight change norm by T when measuring feature learning strength, since longer training brings minimally effect on the weight change norm.
>
> * We would also like to point out that in Section 3.2.1, we used weight change norm $||W_T-W_0||$ as a measure for feature learning strength [2-3]. The weight change norm computes the Frobenius distance between the trained weights and their initialization for each network (within network comparison, trained vs initialized weights). This is distinct from the weight degeneracy, measured as the mean permutation-invariant Frobenius distance across 50 networks at the end of training (across-network comparison). A higher weight change norm from initialization indicates stronger feature learning, but may not necessarily lead to higher weight degeneracy, since the latter depends on the loss landscape geometry. For example, all networks travel far from their initialization, but are clustered close together in the loss landscape after training will have high weight change norm, but low weight degeneracy.
>
> * The reviewer also hinted at an excellent point: the permutation-invariant Frobenius distance used to quantify weight degeneracy can change with respect to the magnitude of the weight matrices being compared. Here, we might be tempted to normalize the weight matrices by the number of training steps, or directly by their magnitude to bring them to the same scale then compute PIF distance. However, with the presence of tanh nonlinearity, scaling the weight magnitude changes the input-output mapping of the networks (e.g. for a network with smaller weight norms, upscaling it may bring the hidden activations to saturation in tanh nonlinearity). In other words, normalizing the weights by their magnitude / training steps distorted the computation they perform. We therefore do not normalize of the weight matrices before computing the weight degeneracy either, since the weight norm is entangled with its computation.
>
> > Q3: What about $l_2$ regularization? I'm assuming it has a similar effect as $l_1$ and low-rank penalties. I think it would be helpful to add regularization experiments in section 3.5 for completeness.
>
> Yes! We do confirm that L2 penalty induces qualitatively same effect on degeneracy as L1 and rank regularization. Here we provide data on the 3BFF task, and will add L2 effect for all tasks to the supplementary material in the camera ready version.
>
> | Regularization Strength | $\lambda$=0 | $\lambda$=1e-5 | $\lambda$=5e-5 | $\lambda$=1e-4 |
> |---|---|---|---|---|
> | Dynamic | 0.122 ± 1.58e-04 | 0.120 ± 1.52e-04 | 0.063 ± 1.20e-04 | 0.050 ± 8.00e-05 |
> | Weight | 9.49e-04 ± 8.66e-07 | 9.25e-04 ± 8.46e-07 | 5.49e-04 ± 1.80e-06 | 3.49e-04 ± 1.43e-06 |
> | Behavior | 1.14e-03 | 8.64e-04 | 9.89e-05 | 4.23e-05 |
>
>
> > Q4: In figure 2B, for the sine-wave generation task, are the colored dots fixed points or samples from trajectories?
>
> We thank the reviewer for noticing this, the colored dots are indeed trajectories of neural activities. We will change the visualization to show the fixed points as points and the trajectories as lines in the revised version of the paper to enhance consistency across figures.
>
> &nbsp;
>
>
> ### Weaknesses
>
> > W1: The behavioral degeneracy is defined as out-of-distribution generalization to longer time sequences (temporal generalization). However, this represents only a narrow slice of possible generalization we might expect networks to exhibit.
>
> We will add temporal generalization in brackets everywhere to clarify the specific form of behavioral degeneracy that we are assessing in our study. Your suggestion to  explore other types of generalization such as response to noise, to input / activation perturbations etc. is an excellent suggestion for future work!
>
> &nbsp;
>
> ### References
> [1] N. Maheswaranathan et al., “Universality and individuality in neural dynamics across recurrent networks,” NeurIPS, 2019.
>
> [2] Y. H. Liu, A. Baratin, J. Cornford, S. Mihalas, E. Shea-Brown, and G. Lajoie, “How connectivity structure shapes rich and lazy learning in neural circuits,” ICLR, 2024.
>
> [3] T. George, G. Lajoie, and A. Baratin, “Lazy vs. hasty: Linearization in deep networks impacts learning schedule based on example difficulty,” TMLR, 2022.

---

> > ### Comment · Reviewer_VYSY · 2025-08-03
> > **Thanks for the reply**
> >
> > - I appreciate adding the phrase [temporal generalization] to specify the type of behavioral degeneracy; I think that sufficiently addresses my concern.
> > - For the weight change, adding some example plots of $\|\| \Delta W \|\|$ during training in the appendix might be illustrative.

---

> ### Author Response · Authors · 2025-08-06
>
> Thanks for the suggestion! We will add example plots of $|| \Delta W ||$ over training to the supplemental material in the revised paper. We thank you again for your insightful review and constructive comments.

---

### Official Review · Reviewer_pYdB · 2025-07-07

**Clarity:** 3
**Significance:** 4
**Originality:** 4
**Rating:** 5
**Confidence:** 4

**Summary:**

This paper investigates solution degeneracy in task-trained RNNs, which occurs when different networks achieve similar task performance (as measured by their training loss) but vary widely in weights, neural dynamics, and behavior.  The authors identify 4 crucial factors (task complexity, learning regime, network size, regularization) that control solution degeneracy in various task-trained RNNs, providing empirical support for the “contravariance principle.” They propose novel metrics to quantify solution degeneracy and show that task complexity and learning regime have contravariant effects (i.e., affect weights and dynamics in opposite ways), while network size and regularization have covariant effects.

**Questions:**

1. Definition of degeneracy: Has the definition of behavioral degeneracy you use in Section 2.3.1 appeared in prior literature? If not, please state that it is novel; if it has, cite the appropriate references.
2. Task motivation: Could you elaborate on why these four tasks were chosen? Do they correspond to known computational functions of brain areas, or are they commonly used RNN benchmarks in neuroscience or machine learning?
3. Path integration claim: On line 127, you say that “After training, the network forms a Euclidean map of the environment in its internal state space.” Can you back this up? Many studies involving RNNs trained on the task of path integration find toroidal neural manifolds in the internal space, similar to grid cells in the mammalian entorhinal cortex. These are non-Euclidean.
4. Clarify the $\phi$ model: In Section 3.3, you introduce a hidden state update involving $\phi$ without context. Is this a new model? How does it tie into your prior experiments?
5. Initialization and $h_0$: How is the initial hidden state $h_0$ set in your experiments?

**Ethical Concerns:**

["NO or VERY MINOR ethics concerns only"]

**Final Justification:**

This is a technically solid paper and the authors have adequately addressed my remaining questions and concerns in their rebuttal. Therefore, I believe it should be accepted, and I am maintaining my score of 5.

**Limitations:**

Yes

**Quality:**

4

**Strengths And Weaknesses:**

**Strengths:**

- Provides empirical validation for the Contravariance Principle.
- Identifies four key factors controlling degeneracy: task complexity, learning regime, network size, and regularization.
- Uses a solid set of degeneracy metrics.
- Well-written and thoughtfully designed set of experiments.

**Weaknesses:**

- Terms like *“learning regime”* and *“feature learning”* are used interchangeably. The paper relies on the Rich vs Lazy regime literature, but this may not be familiar to all readers (especially in neuroscience), and could benefit from clearer terminology introduced earlier.
- In Figure 2, subfigure 2B is not clearly described (presumably PCA of neural activity). It should be clarified in the caption.
- In Section 3.3, $\phi$ is not defined, and a new update rule is introduced without context—unclear how this relates to prior experiments.
- Line 267–268: Inconsistent reasoning—on one hand, the text says that high γ leads to consistent behavior; on the other, it says stronger feature learning leads to less consistent behavior.
- Minor typo on line 160: “compleity”.
- Some sections could benefit from better citations.

---

> ### Author Rebuttal · Authors · 2025-07-30
>
> Thank you for your thoughtful and supportive review! We are glad that you find the paper well-written, and like our empirical validation of the Contravariance Principle, and our systematic analysis of four factors shaping solution degeneracy with solid metrics. Here we address the points you raised in the Questions and Weaknesses section.
>
> ### Questions
>
> > Q1: Has the definition of behavioral degeneracy you use in Section 2.3.1 appeared in prior literature?
>
> To the best of our knowledge, our definition of behavioral degeneracy has not appeared in prior literature. Previous work, such as [1], has reported that image classification models trained from random initializations can have varied performance on out-of-distribution data. However, no explicit definition of behavioral variability or behavioral degeneracy was made there.
>
> > Q2: Could you elaborate on why these four tasks were chosen?
>
> Indeed we picked these tasks by surveying prior work in computational neuroscience and neural computation. Below we list several papers that motivated our choices:
> * N-BFF: pattern recognition / memory retrieval; Hopfield‑type
> attractor networks store binary patterns and retrieve them from partial cues [2,3].
> * Delayed Discrimination: working memory maintenance; dorsolateral prefrontal cortex neurons show direction‑ or stimulus‑specific persistent “delay‑period” firing activities that bridges the sample and test period in delayed‐response / discrimination tasks [4,5].
> * Sine Wave Generation: Central Pattern Generators (CPGs) produce self‑sustaining sinusoidal motor commands [6]; rhythmic and oscillatory activity in the motor cortex during movement [7].
> * Path Integration: Hippocampal place cells and entorhinal grid cells integrate self‑motion to update an internal position estimate when external cues are absent [8].
>
> Furthermore, previous work that developed benchmark neuroscience-relevant task suites have used similar tasks including the N-BFF, Sine Wave Generation, and Delayed Discrimination tasks [9-11].
>
> > Q3: Networks form a map of the environment on Path Integration
>
> We appreciate that the reviewer pointed out that it may not be an Euclidean map. We apologize for any confusion and will remove that from our revised manuscript.
>
> > Q4: Clarify the $\phi$ model
>
> To test feature learning's effect on solution degeneracy, we used a principled network parameterization called muP (maximal update parameterization) to systematically control for feature learning strength. muP enables stable feature learning strength across network widths, even in the infinite-width limit [12]. Line 259-261 in Section 3.3 describes the update rule, and scaling of initialization and learning rate under the muP parameterization. Here, $\phi$ is the tanh nonlinearity, consistent with our discrete-time RNN model introduced in section 2.1. The time constant is 1. In this manuscript, we use the standard discrete-time RNNs due to its practical relevance for task-driven modeling community, while switching to μP to isolate the effect of feature learning and network size in Section 3.3 and 3.4, respectively. In Appendix H, we show a theoretical equivalence between the muP parameterization and the discrete-time RNN setting under proper scaling of the network initialization and update. Moreover, we confirm that the feature learning and network size effects on degeneracy hold qualitatively the same in standard discrete-time RNNs, unless where altering network width induces unstable and lazier learning strength in larger networks in Appendix L-M.
>
> > Q5: How is the initial hidden state $h_0$ set in your experiments?
>
> We initialize our hidden state to be all zeros.
>
> &nbsp;
>
> ### Weaknesses
> > The paper relies on the Rich vs Lazy regime literature, but this may not be familiar to all readers (especially in neuroscience), and could benefit from clearer terminology introduced earlier.
>
> Thank you for your suggestion! We will expand our current introduction on Rich vs Lazy regime in section 3.2., to incorporate more detailed explanation of the terminology, and also discuss their relevance for the neuroscience community [13-15].
>
>
> > In Figure 2, subfigure 2B is not clearly described (presumably PCA of neural activity)
>
> Thanks for catching this! Figure 2B indeed plots the PCA of neural activity in the top principle components' state space. We will clarify this point in the caption of figure 2B.
>
> > Line 267-268: Inconsistent reasoning—on one hand, the text says that high γ leads to consistent behavior; on the other, it says stronger feature learning leads to less consistent behavior.
>
> We clarify that higher $\gamma$, which corresponds to stronger feature learning, leads to more consistent neural dynamics (lower dynamical degeneracy) but less consistent weight structure (higher weight degeneracy).
>
> > Minor typo on line 160: “compleity”
>
> Thanks for noticing it. We will fix the typo.
>
> &nbsp;
> Many thanks again for your careful reading and constructive suggestions! Hope our replies above clarify your questions, and we appreciate the comments that helped us improve the presentation of the paper, which will be incorporated into the revised version of the paper.
>
> &nbsp;
>
>
> ### References
> [1] A. D’Amour et al., “Underspecification presents challenges for credibility in modern machine learning,” JMLR, 2022.
>
> [2] J. J. Hopfield, “Neural networks and physical systems with emergent collective computational abilities,” Proc. Natl. Acad. Sci. USA, vol. 79, no. 8, pp. 2554–2558, 1982.
>
> [3]  I. Jarne, “Exploring flip flop memories and beyond: training recurrent neural networks with key insights,” Frontiers in Systems Neuroscience, 2024.
>
> [4] S. Funahashi, C. J. Bruce, and P. S. Goldman‑Rakic, “Mnemonic coding of visual space in the monkey’s dorsolateral prefrontal cortex,” J. Neurophysiol., vol. 61, no. 2, pp. 331–349, 1989.
>
> [5] P. S. Goldman‑Rakic, “Cellular basis of working memory,” Neuron, vol. 14, no. 3, pp. 477–485, 1995.
>
> [6] E. Marder and D. Bucher, “Central pattern generators and the control of rhythmic movement,” Curr. Biol., vol. 11, no. 23, pp. R986–R996, 2001.
>
> [7] M. M. Churchland et al., “Neural population dynamics during reaching,” Nature, vol. 487, no. 7405, pp. 51–56, 2012.
>
> [8] B. L. McNaughton, F. P. Battaglia, O. Jensen, E. I. Moser, and M.-B. Moser, “Path integration and the neural basis of the ‘cognitive map’,” Nat. Rev. Neurosci., vol. 7, pp. 663–678, 2006.
>
> [9] G. R. Yang, M. R. Joglekar, H. F. Song, W. T. Newsome, and X.‑J. Wang, “Task representations in neural networks trained to perform many cognitive tasks,” Nat. Neurosci., 2019.
>
> [10] M. Khona, S. Chandra, J. J. Ma, and I. R. Fiete, “Winning the Lottery With Neural Connectivity Constraints: Faster Learning Across Cognitive Tasks With Spatially Constrained Sparse RNNs,” Neural Comput., vol. 35, no. 11, 2023, doi: 10.1162/neco_a_01613.
>
> [11] N. Maheswaranathan et al., “Universality and individuality in neural dynamics across recurrent networks,” NeurIPS, 2019.
>
>
> [12] G. Yang and E. J. Hu, “Feature Learning in Infinite-Width Neural Networks,” in Proc. ICML, 2021.
>
> [13] Y. H. Liu, A. Baratin, J. Cornford, S. Mihalas, E. Shea-Brown, and G. Lajoie, “How connectivity structure shapes rich and lazy learning in neural circuits,” ICLR, 2024.
>
> [14] T. Flesch, K. Juechems, T. Dumbalska, A. Saxe, and C. Summerfield, “Orthogonal representations for robust context-dependent task performance in brains and neural networks,” Neuron, vol. 110, no. 7, pp. 1258–1270, 2022.
>
> [15] M. Farrell, S. Recanatesi, and E. Shea-Brown, “From lazy to rich to exclusive task representations in neural networks and neural codes,” Curr. Opin. Neurobiol., vol. 83, p. 102780, 2023.

---

> > ### Comment · Reviewer_pYdB · 2025-08-07
> > **Thanks for your response**
> >
> > The authors have addressed all of my remaining questions and concerns. I am maintaining my score to accept (5).

---

### Decision · Program_Chairs · 2025-09-17

**Decision:**

Accept (spotlight)

**Comment:**

This paper investigates solution degeneracy in task-trained recurrent neural networks (RNNs), a well-documented phenomenon in which different networks achieve similar task performance using different network solutions (i.e., different weights and/or dynamics). This paper identifies 4 crucial factors governing this degeneracy and propose new metrics to quantify it. The problem is timely and important, and is likely to be of interest to a broad audience at NeurIPS. The reviewers were unanimous in their assessment that the paper makes a worthwhile contribution to the literature and should be accepted.  Congratulations! Please be sure to address all reviewer comments and criticisms in the final manuscript.